# Information-Theoretic Disentangled Latent Modeling with Conditional Diffusion for Incomplete Multi-View Clustering

**Wenlan Chen** [1]  **Lu Gao** [1]  **Daoyuan Wang** [1]  **Cheng Liang** [* 2]  **Fei Guo** [* 1]

## Abstract

Incomplete multi-view clustering is challenging due to view missingness and the entanglement of shared semantics with view-specific factors in latent representations. Existing methods often rely on heuristic fusion or direct completion strategies, which suffer from error propagation and unreliable generation under missing views. In this paper, we propose an **I**nformation-guided **D**isentangled latent modeling framework with **C**onditional **D**iffusion for incomplete multi-view clustering (IDCD). Specifically, we first encode each view into a latent representation that is variationally decomposed into a view-wise semantic latent and a view-specific factor. Information-theoretic objectives are introduced to guide the disentanglement of view-wise latents, preserving essential multi-view information while reducing the dependency between semantic and view-specific factors and encouraging cross-view semantic consistency. Besides, we aggregate the semantic latents via a mixture of Wasserstein distributions to obtain a unified global representation, where we impose a Gaussian mixture prior to explicitly couple representation learning with clustering. Based on the learned disentangled latent space, a conditional diffusion model guided by both the global semantic latent and view-specific factors is employed to generate missing views in a consistent manner. Extensive experiments on benchmark datasets demonstrate superior clustering performance and robust missing-view generation compared to state-of-the-art methods.

---

[1]School of Computer Science and Engineering, Central South University, Changsha, China [2]School of Computer Science and Artificial Intelligence, Shandong Normal University, Jinan, China . Correspondence to: Cheng Liang <alcs417@sdnu.edu.cn>, Fei Guo <guofei@csu.edu.cn>.

*Proceedings of the $43^{rd}$ International Conference on Machine Learning*, Seoul, South Korea. PMLR 306, 2026. Copyright 2026 by the author(s).

## 1. Introduction

Multi-view Clustering (MVC) is a core task in unsupervised representation learning that leverages complementary information from multiple data sources to discover a consensus clustering structure (Trosten et al., 2021; Huang et al., 2023; Liang et al., 2025; Yang et al., 2025; Geng et al., 2025). By integrating heterogeneous yet semantically related views, MVC learns representations that better capture the intrinsic data distribution than any single view alone (Lou et al., 2025a; Chen et al., 2025a; Lou et al., 2025b). However, MVC methods assume fully observed views for each instance, which enables direct cross-view alignment, consistency regularization and shared latent representation learning. In practice, multi-view data are often incomplete: some views may be partially observed or entirely missing due to sensor failure, privacy constraints or heterogeneous acquisition processes (Hu & Chen, 2018; Wen et al., 2023; Xu et al., 2024b; Yuan et al., 2025). Such missing views often break cross-modal co-occurrence, weaken semantic alignment, and hinder the exploitation of complementary information, causing standard MVC methods to produce biased representations and degraded clustering performance. These challenges have led to Incomplete Multi-view Clustering (IMVC), which seeks to learn robust clustering structures when only a subset of views is available per instance (Gu et al., 2024; Yu et al., 2024; Xing et al., 2024; Dong et al., 2025). Existing IMVC approaches can generally be divided into non-imputation-based and imputation-based methods, depending on whether missing views are explicitly reconstructed. Non-imputation-based methods perform clustering directly on observed views without explicitly reconstructing missing data, thus avoiding errors introduced by inaccurate imputation. Early studies project incomplete multi-view data into a shared latent space and fuse available views through shallow alignment strategies, such as latent subspace clustering using existing views (Li et al., 2014) and graph-regularized matrix factorization with adaptive view weighting (Wen et al., 2021b). More recent approaches employ deep models to enhance semantic consistency (Tang & Liu, 2022; Chen et al., 2025b). For example, Wu et al. transfer clustering knowledge from complete to incomplete multi-view data through a teacher–student framework (Wu et al., 2025) and Dai et al. learn view-specific representa-

tions with autoencoders and refine consensus features via cross-view contrastive learning (Dai et al., 2025). Generative latent-variable models have also been explored, such as VAE-based joint representation and fusion frameworks (Cai et al., 2024; Gao & Pu, 2025; Xu et al., 2024b). Despite improved modeling capacity, these methods rely solely on partial observations and assume reliable cross-view correspondence or structural consistency, which often break down under high missing rates or severe view imbalance, leading to biased shared representations (Xu et al., 2024a).

Imputation-based methods aim to recover missing views during training to mitigate information loss caused by view absence. Early techniques mainly rely on shallow reconstruction mechanisms, such as kernel-based, graph-based and matrix factorization-based methods (Shudong et al., 2019; Wang et al., 2020). With the development of deep learning, recent methods adopt structure-aware filling strategies, such as prototype- or graph-guided completion with contrastive learning (Wen et al., 2021a; Hong et al., 2025), as well as prediction-based generative models, including GAN-based cross-view prediction (Wang et al., 2021; Xu et al., 2019) and mutual-information-driven auxiliary prediction networks (Lin et al., 2023). More recently, diffusion-based completion models (Wen et al., 2024; Palumbo et al., 2024; Zhang et al., 2025) leverage strong generative capability to synthesize missing views. However, these approaches often depend on strong pairing assumptions and large amounts of aligned data. In addition, these methods often follow a two-stage approach. They do not embed the clustering structure during model training, which prevents them from guaranteeing effective clustering.

To address these challenges, we propose an information-theoretic disentangled latent modeling framework with conditional diffusion for incomplete multi-view clustering. The overall architecture is illustrated in Figure 1. Firstly, the model learns a structured latent space that explicitly separates cross-view shared semantics from view-specific variations through variational latent modeling, establishing a factorized semantic representation that disentangles shared structure from modality-dependent characteristics. Next, semantic information from available views is aggregated at the distribution level through a mixture of Wasserstein barycenters, yielding a unified global representation whose structure aligns with the underlying clustering organization of the latent space. We then introduce information-theoretic regularization to preserve essential multi-view information, control redundancy between latent components and promote cross-view semantic alignment. Lastly, conditioned on structurally disentangled semantic and view-specific factors, a conditional latent diffusion model is employed to recover missing views, which enables controlled modeling of complex conditional distributions and ensures stable generative behavior. Our work makes the following key contributions:

- We introduce a principled latent representation paradigm that disentangles cross-view shared semantics from view-specific variations and regulates their interaction through information-theoretic constraints. Combined with geometry-aware distributional semantic integration, this design enables the formation of coherent and non-redundant global representations that naturally encode clustering structure under arbitrary missing-view patterns.

- We develop a generative mechanism that operates on structurally disentangled latent conditions, allowing controlled modeling of complex conditional distributions and stable recovery of missing views. This design enforces semantic structure during view completion instead of relying on unconstrained feature generation and preserves cross-view semantic coherence.

- Extensive experiments on five benchmark datasets validate the effectiveness of the proposed framework, which consistently outperforms ten state-of-the-art deep incomplete multi-view clustering methods across diverse missing-data scenarios.

## 2. Background

Diffusion models define a forward process that gradually transforms data $\boldsymbol{x}_0$ into noisy representations via a fixed Markov chain (Ho et al., 2020):

$$q(\boldsymbol{x}_t \,|\, \boldsymbol{x}_{t-1}) = \mathcal{N}(\sqrt{1-\beta_t}\,\boldsymbol{x}_{t-1}, \beta_t\mathbf{I}), t = 1, \ldots, T, \quad (1)$$

where $\beta_t$ controls the noise scale at step $t$. This forward process admits a closed-form marginal distribution:

$$q(\boldsymbol{x}_t \,|\, \boldsymbol{x}_0) = \mathcal{N}(\sqrt{\bar{\alpha}_t}\,\boldsymbol{x}_0, (1-\bar{\alpha}_t)\mathbf{I}), \bar{\alpha}_t = \prod_{s=1}^{t}(1-\beta_s), \quad (2)$$

which allows direct sampling of intermediate noisy states from the original data. The generative process inverts this forward chain by sequentially sampling

$$p_\theta(\boldsymbol{x}_{t-1} \,|\, \boldsymbol{x}_t) = \mathcal{N}(\mu_\theta(\boldsymbol{x}_t, t), \boldsymbol{\Sigma}_t), \quad (3)$$

where the mean is parameterized via predicted noise:

$$\mu_\theta(\boldsymbol{x}_t, t) = \frac{1}{\sqrt{\alpha_t}}\Big(\boldsymbol{x}_t - \frac{\beta_t}{\sqrt{1-\bar{\alpha}_t}}\,\epsilon_\theta(\boldsymbol{x}_t, t)\Big), \quad (4)$$

and the covariance is set to $\boldsymbol{\Sigma}_t = \tilde{\beta}_t\mathbf{I}$ with $\tilde{\beta}_t = \frac{1-\bar{\alpha}_{t-1}}{1-\bar{\alpha}_t}\beta_t$, corresponding to the true posterior variance $q(\boldsymbol{x}_{t-1} \,|\, \boldsymbol{x}_t, \boldsymbol{x}_0)$. Training is achieved by maximizing a variational lower bound on $\log p(\boldsymbol{x}_0)$, which reduces under the fixed-variance parameterization to the familiar noise-prediction objective:

$$\mathcal{L}_{\text{simple}} = \mathbb{E}_{t,\boldsymbol{x}_0,\epsilon}\big[\|\epsilon - \epsilon_\theta(\boldsymbol{x}_t, t)\|^2\big]. \quad (5)$$

This objective encourages the network to accurately predict the injected noise at each step, enabling high-fidelity generation from pure Gaussian noise.

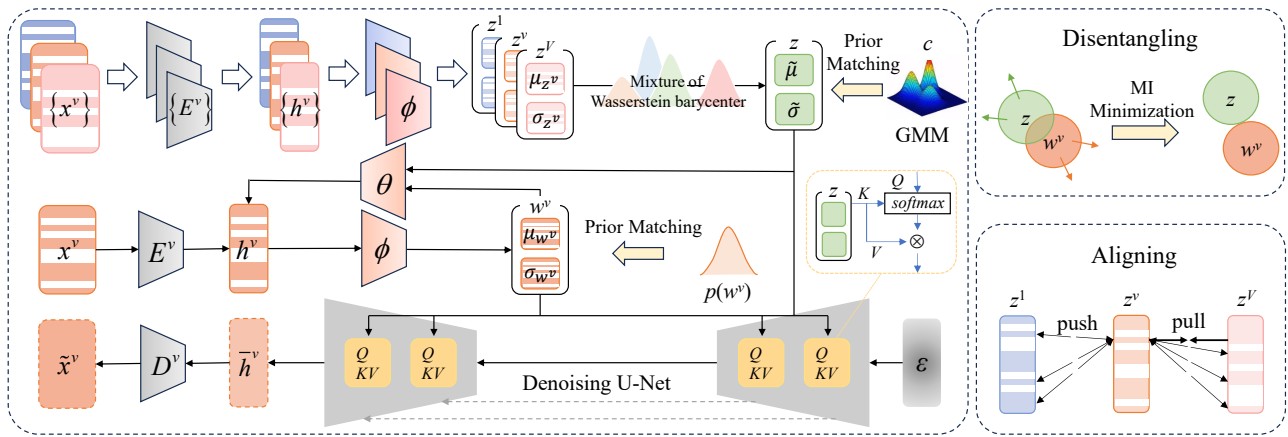

*Figure 1.* Given incomplete multi-view inputs $\{\boldsymbol{x}^v\}_{v=1}^V$, each view is encoded by a view-specific encoder $E^v$ to obtain latent features $\boldsymbol{h}^v$, factorized into shared representations $\boldsymbol{z}^v$ and view-specific variables $\boldsymbol{w}^v$. A Mixture of Wasserstein Barycenter aggregates $\{\boldsymbol{z}^v\}_{v=1}^V$ into a unified representation $\boldsymbol{z}$, regularized by a GMM prior $p(\boldsymbol{z} \mid \boldsymbol{c})$. To achieve disentanglement and alignment, we introduce a variational mutual information objective to separate $\boldsymbol{z}$ from $\boldsymbol{w}^v$, together with contrastive mutual information maximization over $\{\boldsymbol{z}^v\}_{v=1}^V$ for cross-view consistency. For missing-view reconstruction, a conditional denoising U-Net predicts $\bar{\boldsymbol{h}}^v$ from noisy latents with cross-attention, decoded by $D^v$ to reconstruct $\tilde{\boldsymbol{x}}^v$.

## 3. Method

### 3.1. Problem Statement

Let $\mathcal{X} = \{\boldsymbol{X}^v\}_{v=1}^V$ denote an incomplete multi-view dataset with $V$ views. Each view $\boldsymbol{X}^v \in \mathbb{R}^{n_v \times d_v}$ contains $n_v$ observed samples of dimension $d_v$, where $n_v \leq n$ and $n$ is the total number of instances. Let $\boldsymbol{x}_i^v \in \mathbb{R}^{d_v}$ denote the $i$-th sample in the $v$-th view. For brevity, the subscript $i$ is omitted when the context is clear. Due to missing modalities, the multi-view representation of an instance is expressed as $\{\boldsymbol{x}^v\}_{v=1}^V$, which may be partially observed. We introduce binary indicator vectors $\boldsymbol{g}^v \in \{0,1\}^n$, where $g_i^v = 1$ indicates that instance $i$ is observed in view $v$ and $g_i^v = 0$ otherwise. The goal is to partition the $n$ instances into $K$ clusters, such that the learned representation captures the underlying semantic structure despite incomplete observations. All symbols and their meanings are summarized in Table 3.

### 3.2. Disentangled Multiview Latent Conditioning for Diffusion Models

Inspired by latent diffusion models (Rombach et al., 2022), we also perform diffusion in the latent space. In essence, data from different modalities often exhibit heterogeneous scales and geometric structures, making diffusion in the raw observation space inefficient and hard to optimize. Therefore, we first map each view to a compact latent representation via an encoder $\boldsymbol{h}^v = E^v(\boldsymbol{x}^v)$, which yields a normalized and structured space better suited for diffusion modeling.

### 3.2.1. VARIATIONAL INFERENCE AND DIFFUSION FORWARD PROCESS.

Existing diffusion-based completion methods often rely on strong data alignment, require large amounts of paired data and do not embed clustering structure during training. Motivated by these limitations, we formulate the inference procedure of the model under a unified latent-variable perspective, whose overall variational posterior factorizes as follows:

$$
\begin{aligned}
&q(\boldsymbol{z}, \boldsymbol{c}, \{\boldsymbol{w}^v\}, \{\boldsymbol{h}_{1:T}^v\} \mid \{\boldsymbol{h}_0^v\}) \\
&= q(\boldsymbol{z} \mid \{\boldsymbol{h}_0^v\}) q(\boldsymbol{c} \mid \boldsymbol{z}) \prod_{v=1}^V q(\boldsymbol{h}_{1:T}^v \mid \boldsymbol{h}_0^v) q(\boldsymbol{w}^v \mid \boldsymbol{h}_0^v)
\end{aligned} \quad (6)
$$

where $q(\boldsymbol{z} \mid \{\boldsymbol{h}_0^v\})$, $q(\boldsymbol{c} \mid \boldsymbol{z})$ and $q(\boldsymbol{w}^v \mid \boldsymbol{h}_0^v)$ constitute the variational inference component that learns structured latent variables, while $q(\boldsymbol{h}_{1:T}^v \mid \boldsymbol{h}_0^v)$ corresponds to the diffusion forward process that models the stochastic evolution of latent representations.

**Variational Inference.** In the following, we first detail the variational inference component, which infers the shared semantic latent $\boldsymbol{z}$, the discrete category latent $\boldsymbol{c}$ and view-specific factors $\boldsymbol{w}^v$. Given that the true posterior $p(\boldsymbol{z}, \boldsymbol{c}, \{\boldsymbol{w}^v\} \mid \{\boldsymbol{h}_0^v\})$ is intractable, we employ a variational approximation $q_\phi(\boldsymbol{z}, \boldsymbol{c}, \{\boldsymbol{w}^v\} \mid \{\boldsymbol{h}_0^v\})$ capable of accommodating arbitrary missing-view configurations. Specifically, each view-specific factor is modeled independently as

$$
q_\phi(\boldsymbol{w}^v \mid \boldsymbol{h}_0^v) = \mathcal{N}\big(\boldsymbol{w}^v \mid \boldsymbol{\mu}_{\boldsymbol{w}^v}, \boldsymbol{\sigma}_{\boldsymbol{w}^v}^2 \mathbf{I}\big) \quad (7)
$$

where $[\boldsymbol{\mu}_{\boldsymbol{w}^v}, \boldsymbol{\sigma}_{\boldsymbol{w}^v}^2] = \mathcal{H}_w^v(\boldsymbol{h}_0^v)$ and $\mathcal{H}_w^v(\cdot)$ is a view-specific encoder that maps the encoded observation $\boldsymbol{h}_0^v$ to the mean and variance of the Gaussian. $\boldsymbol{w}^v$ captures modality-dependent variations that are not shared across views, such

as acquisition-specific noise, style or modality-specific distortions.

To obtain a reliable shared semantic latent representation $z$, we further project $h_0^v$ into $z^v$ and model the distribution of $z^v$ as $q_\phi(z^v \mid h_0^v) = \mathcal{N}(z^v \mid \mu_{z^v}, \sigma_{z^v}^2 \mathbf{I}), [\mu_{z^v}, \sigma_{z^v}^2] = \mathcal{H}_z^v(h_0^v)$. Since each view is processed by an independently parameterized encoder operating solely on its corresponding observation, the variational posterior can be factorized across views as:

$$q_\phi(z \mid \{h_0^v\}_{v=1}^V) = \int \cdots \int q_\phi(z \mid \{z^v\}) \prod_{v=1}^V q_\phi(z^v \mid h_0^v) dz^1 \cdots dz^V$$

$$= \mathbb{E}_{q_\phi^1(z^1 \mid h_0^1)} \cdots \mathbb{E}_{q_\phi^V(z^V \mid h_0^V)} [q_\phi(z \mid \{z^v\})],$$
(8)

where $q_\phi(z \mid \{z^v\})$ represents the conditional distribution that aggregates the modality-specific semantic latents into the shared semantic latent and captures shared semantic information across views. To better approximate the posterior $q_\phi(z \mid \{z^v\}_{v=1}^V)$ and account for missing-view scenarios, we adopt a Mixture of Wasserstein Barycenters (MWB) to fuse the view-specific semantic latents. Unlike naive averaging or product-based aggregation, MWB preserves the geometric structure of the latent space by leveraging the 2-Wasserstein distance, which provides a balanced and structure-aware integration of modality-specific posteriors. Formally, for each observation, we first compute the Wasserstein barycenter over the available modality subset $Z_m$:

$$q_{\text{WB}} = \arg \min_q \sum_{v \in Z_m} \lambda_v \mathcal{W}_2^2(q_\phi(z^v \mid h_0^v), q), \quad (9)$$

where $\lambda_v = 1/|Z_m|$ are equal weights assigned to the participating modalities and $\mathcal{W}_2^2$ denotes the squared 2-Wasserstein distance. This procedure yields a central distribution that is geometrically closest to all available modality-specific posteriors, thereby aggregating cross-view semantic information while preserving the intrinsic structure of the latent space. Since each modality-specific posterior $q_\phi(z^v \mid h_0^v)$ is defined as an isotropic Gaussian, the barycenter admits a closed-form solution by Theorem A.1:

$$\tilde{\mu} = \sum_{v \in Z_m} \lambda_v \mu_{z^v}, \tilde{\sigma} = \sum_{v \in Z_m} \lambda_v \sigma_{z^v}. \quad (10)$$

To handle arbitrary missing-view patterns, we extend this aggregation over all possible view subsets $Z_m \in \mathcal{Z}$, where $\mathcal{Z} = \mathcal{P}(\{1, \ldots, V\})$ and $\mathcal{P}(\cdot)$ denotes the powerset. The final shared posterior is then obtained via an outer KL projection:

$$q_\phi(z \mid \{h_0^v\}_{v=1}^V) = \arg \min_q \sum_{Z_m \in \mathcal{Z}} \lambda_m KL(q_{\text{WB}} \| q), \quad (11)$$

where $\lambda_m$ are normalized weights over modality subsets. In practice, one subset $Z_m$ is randomly sampled per training

iteration and $q_\phi(z \mid \{h_0^v\})$ is approximated with the corresponding barycenter $q_{\text{WB}}$, which provides an efficient and scalable approximation.

Finally, the discrete category latent $c$ is inferred conditioned on the aggregated shared latent $z$ as

$$q(c \mid z) = p(c \mid z) = \frac{p(c)\, p(z \mid c)}{\sum_{c'} p(c')\, p(z \mid c')}, \quad (12)$$

which ensures that the variational posterior respects the generative assumptions while remaining fully differentiable for end-to-end training.

**Diffusion Forward Process.** Subsequently, we describe the diffusion forward process, which defines the stochastic trajectories of latent representations and forms the basis of the conditional latent diffusion model. For each view, the forward diffusion is applied to the encoded representations $\{h_0^v\}$ according to a variance-preserving Markovian process:

$$q(h_t^v \mid h_{t-1}^v) = \mathcal{N}\left(h_t^v; \sqrt{1 - \beta_t}\, h_{t-1}^v, \beta_t \mathbf{I}\right), t = 1, \ldots, T,$$
(13)

with the corresponding closed-form marginal

$$q(h_t^v \mid h_0^v) = \mathcal{N}\left(h_t^v; \sqrt{\bar{\alpha}_t}\, h_0^v, (1 - \bar{\alpha}_t)\mathbf{I}\right), \bar{\alpha}_t = \prod_{s=1}^t (1 - \beta_s).$$
(14)

This formulation explicitly defines the latent trajectories under the forward diffusion process.

### 3.2.2. GENERATIVE MODELING OF LATENT STRUCTURE.

The above formulation describes the inference perspective of the model. We present the corresponding generative process, summarized as follows:

$$p_\theta(\{h_{0:T}^v\}_{v=1}^V, z, \{w^v\}_{v=1}^V, c)$$

$$= p(c)\, p(z \mid c) \prod_{v=1}^V p(w^v)\, p(h_T^v) \prod_{t=1}^T p_\theta(h_{t-1}^v \mid h_t^v, z, w^v).$$
(15)

This joint factorization consists of two tightly coupled components. The terms $p(c)$, $p(z \mid c)$ and $\prod_{v=1}^V p(w^v)$ define the structured latent prior, which corresponds to the category, semantic and view-specific variables learned through the variational inference module. In contrast, the terms $p(h_T^v)$ and $\prod_{t=1}^T p_\theta(h_{t-1}^v \mid h_t^v, z, w^v)$ constitute the conditional diffusion generative process, which progressively reconstructs view representations conditioned on the latent variables through a reverse diffusion chain.

**Hierarchical Latent Prior.** We formalize a hierarchical latent-variable model to characterize the semantic structure

underlying multi-view data, which serves as the probabilistic conditioning space for the diffusion process. Let the hierarchical semantic prior be defined as

$$
\begin{aligned}
p(\boldsymbol{c}) &= \mathrm{Cat}(\boldsymbol{\pi}), \\
p(\boldsymbol{z} \mid \boldsymbol{c} = k) &= \mathcal{N}(\boldsymbol{\mu}_k, \boldsymbol{\sigma}_k^2 \mathbf{I}), \\
p(\boldsymbol{w}^v) &= \mathcal{N}(\mathbf{0}, \mathbf{I}),
\end{aligned}
\tag{16}
$$

where $\boldsymbol{c} \in \{1, \ldots, K\}$ is a discrete latent variable representing high-level semantic categories with prior probabilities $\boldsymbol{\pi}$, satisfying $\sum_{k=1}^{K} \pi_k = 1$ and $\pi_k \geq 0$. Conditioned on $\boldsymbol{c}$, the continuous latent variable $\boldsymbol{z} \in \mathbb{R}^d$ models intra-category semantic variations through a Gaussian component with cluster-specific mean $\boldsymbol{\mu}_k$ and isotropic variance $\sigma_k^2 \mathbf{I}$, which together induce a Gaussian mixture prior over the semantic latent space. $p(\boldsymbol{w}^v)$ is a standard Gaussian prior over view-specific factors, used to constrain modality-dependent variations while providing regularization during both generation and inference. This hierarchical construction decomposes the generative factors into three components: global semantic identity $\boldsymbol{c}$, category-consistent semantic latent $\boldsymbol{z}$, and view-specific variations $\boldsymbol{w}^v$. Such a structured prior promotes cross-view semantic consistency through the shared latent $\boldsymbol{z}$, while explicitly modeling modality-dependent variability via $\boldsymbol{w}^v$.

**Conditional Diffusion Generative Process.** Crucially, this latent-variable formulation defines a semantically structured conditioning space for diffusion-based generation. Rather than conditioning diffusion on arbitrary feature representations, the generative process operates over latent variables endowed with explicit probabilistic semantics and disentanglement properties. The shared variable $\boldsymbol{z}$ enforces cross-view semantic coherence as a global anchor, while $\boldsymbol{w}^v$ enables view-adaptive generation. For each view $v$, generation begins from Gaussian noise $p(\boldsymbol{h}_T^v) \sim \mathcal{N}(0, I)$ and proceeds via a Markovian denoising chain

$$
p_\theta(\boldsymbol{h}_{t-1}^v \mid \boldsymbol{h}_t^v, \boldsymbol{z}, \boldsymbol{w}^v) = \mathcal{N}(\mu_\theta(\boldsymbol{h}_t^v, t, \boldsymbol{z}, \boldsymbol{w}^v), \boldsymbol{\Sigma}_t), \quad (17)
$$

where the reverse diffusion process is modulated by both the shared semantic latent $\boldsymbol{z}$ and the view-specific factor $\boldsymbol{w}^v$. Typically, $\boldsymbol{\Sigma}_t = \tilde{\beta}_t \mathbf{I}$, $\tilde{\beta}_t = \frac{1 - \bar{\alpha}_{t-1}}{1 - \bar{\alpha}_t} \beta_t$. The mean can be expressed in terms of the predicted noise as:

$$
\mu_\theta(\boldsymbol{h}_t^v, t, \boldsymbol{z}, \boldsymbol{w}^v) = \frac{1}{\sqrt{\alpha_t}} \left( \boldsymbol{h}_t^v - \frac{\beta_t}{\sqrt{1 - \bar{\alpha}_t}} \epsilon_\theta(\boldsymbol{h}_t^v, t, \boldsymbol{z}, \boldsymbol{w}^v) \right),
\tag{18}
$$

where $\epsilon_\theta(\cdot)$ predicts the noise term used to parameterize the reverse diffusion transition. The noise predictor is implemented as a U-Net backbone with hierarchical feature processing, where latent semantics are incorporated via adaptive group normalization and cross-attention layers (details in Appendix A.4).

### 3.2.3. EVIDENCE LOWER BOUND OBJECTIVE

From the generative model and the corresponding variational inference scheme, we obtain the following evidence lower bound (ELBO) objective:

$$
\log p\left(\{\boldsymbol{h}_0^v\}\right) \geq -\mathcal{L}_{\mathrm{diff}} - \mathcal{L}_{\mathrm{KL}} - \mathcal{L}_{\mathrm{c}} = -\mathcal{L}_{\mathrm{ELBO}}, \quad (19)
$$

where

$$
\begin{aligned}
\mathcal{L}_{\mathrm{diff}} &= -\sum_{v=1}^{V} \mathbb{E}_{q(\boldsymbol{h}_1^v \mid \boldsymbol{h}_0^v)} \Big[ \mathbb{E}_{q_{\boldsymbol{z}} q_{\boldsymbol{w}^v}} \left[ \log p(\boldsymbol{h}_0^v \mid \boldsymbol{h}_1^v, \boldsymbol{z}, \boldsymbol{w}^v) \right] \Big] + \\
&\sum_{v=1}^{V} \sum_{t=2}^{T} \mathbb{E}_{q_t^v} \Big[ \mathbb{E}_{q_{\boldsymbol{z}} q_{\boldsymbol{w}^v}} KL\left(q(\boldsymbol{h}_{t-1}^v \mid \boldsymbol{h}_t^v, \boldsymbol{h}_0^v) \| p(\boldsymbol{h}_{t-1}^v \mid \boldsymbol{h}_t^v, \boldsymbol{z}, \boldsymbol{w}^v)\right) \Big] \\
&+ \sum_{v=1}^{V} KL\left(q(\boldsymbol{h}_T^v \mid \boldsymbol{h}_0^v) \| p(\boldsymbol{h}_T^v)\right),
\end{aligned}
\tag{20}
$$

$$
\mathcal{L}_{\mathrm{KL}} = \sum_{v=1}^{V} KL(q_{\boldsymbol{w}^v} \| p(\boldsymbol{w}^v)) + \mathbb{E}_{q(\boldsymbol{c} \mid \boldsymbol{z})} KL(q_{\boldsymbol{z}} \| p(\boldsymbol{z} \mid \boldsymbol{c})),
\tag{21}
$$

$$
\mathcal{L}_{\mathrm{c}} = \mathbb{E}_{q_{\boldsymbol{z}}} KL(q(\boldsymbol{c} \mid \boldsymbol{z}) \| p(\boldsymbol{c})),
\tag{22}
$$

where $q_{\boldsymbol{z}} \triangleq q(\boldsymbol{z} \mid \{\boldsymbol{h}_0^v\}_{v=1}^V), q_{\boldsymbol{w}^v} \triangleq q(\boldsymbol{w}^v \mid \boldsymbol{h}_0^v), q_t^v \triangleq q(\boldsymbol{h}_t^v \mid \boldsymbol{h}_0^v)$. The detailed derivation of the above variational lower bound is provided in the Appendix A.5. The objective consists of three components. The first component $\mathcal{L}_{\mathrm{diff}}$ includes the reconstruction likelihood under the conditional diffusion model, which ensures that the denoised latent representations remain consistent with both the shared semantic variable $\boldsymbol{z}$ and the view-specific factor $\boldsymbol{w}^v$. The KL terms associated with $\boldsymbol{h}_t^v$ regularize the latent trajectories by aligning the forward noising distribution with the learned reverse model. The KL term at the terminal step $t = T$ does not affect optimization, since the forward process yields $q(\boldsymbol{h}_T^v \mid \boldsymbol{h}_0^v) = \mathcal{N}(0, \mathbf{I})$, which exactly matches the prior $p(\boldsymbol{h}_T^v) = \mathcal{N}(0, \mathbf{I})$. The second component, $\mathcal{L}_{\mathrm{KL}}$ enforces hierarchical prior constraints on the shared semantic latent $\boldsymbol{z}$ and the view-specific factors $\boldsymbol{w}^v$. The third component $\mathcal{L}_{\mathrm{c}}$ aligns the inferred cluster posterior $q(\boldsymbol{c} \mid \boldsymbol{z})$ with the prior $p(\boldsymbol{c})$, which ensures that the latent categorical variables reflect the intended semantic structure. Together, these components define the overall ELBO objective $\mathcal{L}_{\mathrm{ELBO}} = \mathcal{L}_{\mathrm{diff}} + \mathcal{L}_{\mathrm{KL}} + \mathcal{L}_{\mathrm{c}}$ and guide the joint learning of shared semantics, cluster assignments and view-specific factors.

### 3.3. Information-Theoretic Regularization for Latent Semantics

While the ELBO provides a principled variational framework for learning latent-variable generative models, it primarily focuses on approximating the marginal likelihood of the observed data. As such, relying solely on the ELBO does not explicitly enforce desirable structural properties in the

latent space. In particular, it cannot guarantee that the latent representations are semantically consistent across multiple views or that shared and view-specific factors remain disentangled. Intuitively, while Definition A.2 ensures that the shared and view-specific latents encode non-overlapping generative factors and we focus on functional disentanglement rather than strict identifiability, Definition A.3 guarantees that together they capture all essential information from the multi-view observations. To operationalize these properties, we introduce the following mutual information objective:

$$\mathcal{L}_{\text{dis}} = I(\{\boldsymbol{h}_0^v\}; \boldsymbol{z}, \{\boldsymbol{w}^v\}) - I(\boldsymbol{z}; \{\boldsymbol{w}^v\}), \quad (23)$$

where $I(\{\boldsymbol{h}^v\}; \boldsymbol{z}, \{\boldsymbol{w}^v\})$ encourages the latent variables to retain maximal information from the observed multi-view features, ensuring that global semantics and view-specific factors jointly reflect the underlying data. The term $I(\boldsymbol{z}; \{\boldsymbol{w}^v\})$ penalizes redundant dependency between shared and view-specific latents, promoting disentanglement and factorization of the representation. According to Theorem 3.1, these mutual information terms can be lower-bounded by tractable expectations over KL divergences and conditional log-likelihoods.

**Theorem 3.1** (Structural Information Bound). *By introducing variational distributions $q(\boldsymbol{z} \mid \{\boldsymbol{h}^v\})$ and $q(\boldsymbol{w}^v \mid \boldsymbol{h}^v)$, along with a clustering prior $p(\boldsymbol{z} \mid \boldsymbol{c})$, the mutual information objective $\mathcal{L}_{\text{dis}}$ can be lower-bounded as follows:*

$$\mathcal{L}_{\text{dis}} \geq -\mathcal{L}_{\text{rec}} - \mathcal{L}_{\text{KL}}. \quad (24)$$

*where*

$$\mathcal{L}_{\text{rec}} = -2\mathbb{E}_{q(\{\boldsymbol{h}_0^v\})}\mathbb{E}_{q(\boldsymbol{z}, \{\boldsymbol{w}^v\}|\{\boldsymbol{h}_0^v\})}\left[\log p(\{\boldsymbol{h}_0^v\} \mid \boldsymbol{z}, \{\boldsymbol{w}^v\})\right] \quad (25)$$
*corresponds to the reconstruction likelihood for semantic sufficiency and $\mathcal{L}_{\text{KL}}$ combines the clustering-based alignment for $\boldsymbol{z}$ and the KL penalties for view-specific factors $\boldsymbol{w}^v$, enforcing structured semantic and factor disentanglement.*

*Proof.* See Appendix A.7 for the detailed derivation and expansion of the variational bounds. □

While Theorem 3.1 ensures the disentanglement and semantic sufficiency within each view's latent space, it does not explicitly enforce cross-view alignment. To further strengthen the coherence of the shared latent factors, we introduce a contrastive objective aimed at maximizing the mutual information between the shared representations of different views, denoted as $\sum_{v=1}^{V} \sum_{l \neq v} I(\boldsymbol{z}^v; \boldsymbol{z}^l)$.

**Theorem 3.2** (Contrastive Alignment Lower Bound). *Let $\boldsymbol{z}_i^v$ denote the shared latent representation of the $i$-th data instance under view $v$. The cross-view mutual information admits the following contrastive lower bound:*

$$\sum_{v=1}^{V} \sum_{l \neq v} I(\boldsymbol{z}^v; \boldsymbol{z}^l) \geq \sum_{v=1}^{V} \sum_{l \neq v} \mathbb{E}_i\left[\log \frac{\exp(s(\boldsymbol{z}_i^v, \boldsymbol{z}_i^l)/\tau)}{\sum_{j=1}^{N} \exp(s(\boldsymbol{z}_i^v, \boldsymbol{z}_j^l)/\tau)}\right]$$
$$= -\mathcal{L}_{\text{InfoNCE}}, \quad (26)$$

*where $(\boldsymbol{z}_i^v, \boldsymbol{z}_i^l)$ forms a positive pair from the same instance $i$, $\{\boldsymbol{z}_j^l\}_{j=1}^{N}$ are negative samples from other instances in the minibatch, $s(\cdot, \cdot)$ is a similarity function and $\tau > 0$ is a temperature parameter.*

*Proof.* The full proof is deferred to Appendix A.8. □

Overall, Eq. (24) enforces semantic sufficiency and disentanglement within the latent space and Eq. (26) promotes cross-view semantic alignment. Together, they regularize the latent space to be informative, factorized and semantically consistent across views.

### 3.4. Overall Loss Function

Integrating the diffusion-based variational bound with the structural and contrastive objectives, the overall training loss is

$$\mathcal{L}_{\text{overall}} = \mathcal{L}_{\text{diff}} + \alpha\mathcal{L}_{\text{rec}} + \beta\mathcal{L}_{\text{KL}} + \mathcal{L}_{\text{c}} + \gamma\mathcal{L}_{\text{InfoNCE}} \quad (27)$$

Here, the hyperparameters $\alpha$, $\beta$ and $\gamma$ balance the reconstruction-enhancement term, the hierarchical prior regularization strength and the cross-view mutual-information regularizer, respectively. Based on Lemma A.4 and Theorem A.5, we obtain the following chain of inequalities: $\log p(\{\boldsymbol{h}_0^v\}) \geq -\mathcal{L}_{\text{ELBO}} \geq -\mathcal{L}_{\text{overall}}$. This shows that the proposed objective remains a valid lower bound of the data log-likelihood while incorporating structured semantic regularization. Hence, generative modeling and cross-view semantic alignment are optimized under a unified variational framework rather than through heuristic auxiliary losses. The overall optimization procedure is summarized in Algorithm 1.

## 4. Experiments

### 4.1. Experiment Setup

**Datasets, Baselines and Metrics.** We evaluate the proposed method on five widely used benchmark datasets for multi-view learning, including CUB (Wah et al., 2011), Hand-Written (van Breukelen et al., 1998), LandUse_21 (Yang & Newsam, 2010), CCV (Jiang et al., 2011), Reuters (Lewis et al., 2004) and CARL (Espinosa-Duró, 2013). Detailed information about the datasets is provided in the Appendix B.1. We compare the proposed method with nine state-of-the-art approaches for incomplete multi-view clustering,

*Table 1.* Clustering performance across five datasets under four different missing rates. The best results are highlighted in bold, while the second-best results are underlined.

| | Missing Rates | 0.1 | | | 0.3 | | | 0.5 | | | 0.7 | | |
|---|---|---|---|---|---|---|---|---|---|---|---|---|---|
| | Metrics | ACC | NMI | ARI | ACC | NMI | ARI | ACC | NMI | ARI | ACC | NMI | ARI |
| **CUB** | DIMVC | 66.03 | 61.70 | 48.96 | 57.20 | 56.00 | 41.25 | 60.65 | 55.75 | 42.81 | 56.08 | 51.07 | 36.65 |
| | DSIMVC | 72.93 | 67.82 | 55.89 | 66.83 | 61.78 | 47.71 | 68.37 | 61.55 | 48.21 | _67.33_ | 59.89 | _46.31_ |
| | APADC | 61.93 | 63.79 | 43.55 | 64.23 | 63.28 | 46.24 | 59.70 | 62.17 | 44.11 | 50.33 | 54.49 | 35.33 |
| | CPSPAN | 58.77 | 62.27 | 45.35 | 61.30 | 64.21 | 48.93 | 60.07 | 64.18 | 46.42 | 58.60 | _62.16_ | 45.23 |
| | DVIMC | 44.53 | 41.83 | 23.78 | 43.37 | 45.18 | 28.29 | 39.57 | 34.39 | 20.52 | 39.47 | 36.71 | 22.41 |
| | RPCIC | 71.37 | 70.50 | 57.24 | 67.50 | 67.60 | 54.28 | 59.43 | 60.98 | 44.52 | 58.53 | 61.14 | 45.14 |
| | DCG | _74.57_ | _72.11_ | 59.77 | _71.38_ | _67.82_ | _55.70_ | _68.98_ | _66.30_ | _53.33_ | 62.87 | 60.97 | 45.83 |
| | BURG | 71.50 | 71.60 | _60.48_ | 58.00 | 61.85 | 45.86 | 57.33 | 58.89 | 45.12 | 53.83 | 56.58 | 45.12 |
| | DIMVC-HIA | 73.57 | 69.31 | 57.29 | 70.70 | 64.36 | 52.19 | 62.23 | 56.83 | 43.44 | 53.10 | 48.10 | 33.58 |
| | IDCD | **82.00** | **79.31** | **69.21** | **77.33** | **74.10** | **63.22** | **74.84** | **68.85** | **57.43** | **71.83** | **65.13** | **52.92** |
| **HandWritten** | DIMVC | 89.13 | 80.06 | 77.96 | 85.24 | 74.67 | 70.83 | 82.76 | 71.17 | 66.81 | 79.66 | 68.94 | 63.12 |
| | DSIMVC | 81.27 | 79.47 | 71.59 | 81.82 | 80.27 | 73.36 | 81.39 | 79.23 | 71.88 | 77.38 | 74.80 | 66.84 |
| | APADC | 79.45 | 79.19 | 70.46 | 66.35 | 73.01 | 50.73 | 79.35 | 81.55 | 72.60 | 57.03 | 65.26 | 51.90 |
| | CPSPAN | 80.30 | 78.43 | 71.84 | 79.80 | 79.08 | 72.28 | 84.64 | 80.23 | 75.34 | 83.90 | 80.60 | _75.25_ |
| | DVIMC | 87.89 | 83.51 | 79.66 | 85.36 | _82.82_ | _78.50_ | _85.01_ | _82.96_ | _78.50_ | 81.66 | _80.78_ | 74.85 |
| | RPCIC | _90.86_ | 83.15 | _81.15_ | _87.25_ | 80.91 | 76.87 | 82.36 | 76.93 | 71.17 | 77.59 | 74.74 | 67.01 |
| | DCG | 82.75 | 82.63 | 74.88 | 82.70 | 80.54 | 73.96 | 80.80 | 76.21 | 70.45 | 79.75 | 74.00 | 67.54 |
| | BURG | 86.12 | _84.24_ | 79.16 | 76.75 | 75.69 | 64.72 | 72.82 | 72.75 | 60.16 | 68.70 | 68.03 | 49.84 |
| | DIMVC-HIA | 88.17 | 83.84 | 77.38 | 84.98 | 79.00 | 73.50 | 79.38 | 73.40 | 66.31 | _84.25_ | 77.05 | 72.54 |
| | IDCD | **97.17** | **93.56** | **93.83** | **94.90** | **89.12** | **89.11** | **94.48** | **88.50** | **88.23** | **92.73** | **85.34** | **84.69** |
| **LandUse-21** | DIMVC | 23.29 | 28.85 | 10.12 | 23.13 | 27.61 | 9.61 | 20.44 | 26.39 | 8.07 | 20.62 | 25.71 | 8.15 |
| | DSIMVC | 17.14 | 16.74 | 9.02 | 17.05 | 16.85 | 9.50 | 16.91 | 16.21 | 8.63 | 16.48 | 15.84 | 8.01 |
| | APADC | 24.11 | 28.84 | 11.03 | 23.79 | 27.53 | 10.66 | 23.25 | 13.58 | 10.62 | 21.69 | 24.06 | 8.62 |
| | CPSPAN | 20.86 | 30.26 | 8.55 | 23.19 | 30.47 | 10.83 | _24.90_ | _31.30_ | **13.26** | _23.10_ | **30.71** | _10.36_ |
| | DVIMC | _27.69_ | **35.93** | _14.69_ | _25.63_ | **33.64** | _13.01_ | 22.82 | 30.75 | 10.98 | 19.83 | 26.12 | 7.47 |
| | RPCIC | 21.27 | 26.23 | 8.86 | 21.69 | 26.95 | 8.97 | 17.95 | 23.86 | 6.33 | 18.57 | 24.50 | 6.98 |
| | DCG | 23.88 | 25.89 | 9.77 | 22.86 | 25.37 | 9.47 | 23.75 | 25.58 | 9.63 | 22.79 | 24.12 | 8.86 |
| | BURG | 21.10 | 27.42 | 8.80 | 19.70 | 25.56 | 7.00 | 17.98 | 22.06 | 5.38 | 15.71 | 20.08 | 4.23 |
| | DIMVC-HIA | 19.23 | 21.15 | 6.24 | 21.45 | 23.45 | 7.60 | 22.02 | 24.13 | 8.05 | 21.64 | 23.59 | 7.95 |
| | IDCD | **27.90** | _35.66_ | **15.08** | **27.40** | _33.61_ | **14.02** | **26.65** | **32.61** | _13.08_ | **24.70** | _29.94_ | **11.41** |
| **CCV** | DIMVC | 22.09 | 20.66 | 8.30 | 19.72 | 17.40 | 6.72 | 17.96 | 16.32 | 5.98 | 16.98 | 13.18 | 4.98 |
| | DSIMVC | 19.74 | 18.02 | 7.73 | 18.71 | 16.34 | 6.75 | 19.61 | 16.59 | 6.89 | 18.59 | 15.02 | 6.33 |
| | APADC | 15.19 | 14.31 | 4.71 | 14.59 | 13.15 | 4.45 | 12.31 | 8.43 | 2.53 | 10.66 | 5.24 | 1.28 |
| | CPSPAN | 14.82 | 14.99 | 4.66 | 18.80 | 16.18 | 5.90 | 17.53 | 15.78 | 5.78 | 17.17 | 14.91 | 5.39 |
| | DVIMC | 17.22 | 11.78 | 3.61 | 16.15 | 11.87 | 3.12 | 14.69 | 10.25 | 1.22 | 14.81 | 8.45 | 1.60 |
| | RPCIC | 19.47 | 18.60 | 6.93 | 14.51 | 10.78 | 3.58 | 13.21 | 9.26 | 2.97 | 14.44 | 10.02 | 3.49 |
| | DCG | 19.52 | 17.13 | 7.10 | 17.50 | 15.11 | 5.75 | 18.25 | 14.10 | 6.13 | 17.53 | 12.89 | 5.44 |
| | BURG | 14.66 | 11.12 | 3.57 | 13.85 | 10.03 | 2.80 | 13.02 | 8.98 | 2.19 | 12.83 | 8.19 | 2.16 |
| | DIMVC-HIA | _23.93_ | **22.22** | _8.97_ | _23.28_ | _20.46_ | _8.23_ | _21.08_ | _18.85_ | _7.06_ | _21.24_ | _16.91_ | _7.62_ |
| | IDCD | **24.15** | _21.62_ | **9.22** | **25.39** | **21.45** | **9.47** | **23.12** | **19.78** | **8.31** | **23.01** | **18.86** | **8.09** |
| **Reuters** | DIMVC | 48.83 | 28.94 | 25.78 | 50.54 | 29.86 | 26.89 | 48.51 | 27.29 | 24.74 | 46.94 | 25.79 | 23.24 |
| | DSIMVC | _51.26_ | **35.56** | 28.21 | _51.33_ | _34.88_ | 26.61 | 50.78 | **36.85** | _28.27_ | 47.12 | _33.57_ | 25.51 |
| | APADC | 41.33 | 20.96 | 8.59 | 43.43 | 26.76 | 10.64 | 46.67 | 28.85 | 12.79 | 45.77 | 25.05 | 13.62 |
| | CPSPAN | 38.35 | 14.35 | 10.94 | 38.51 | 13.11 | 10.47 | 38.21 | 11.80 | 11.30 | 37.86 | 12.03 | 10.16 |
| | DVIMC | 44.06 | 16.08 | 15.21 | 43.06 | 10.84 | 11.77 | 35.37 | 5.14 | 4.98 | 32.18 | 3.02 | 3.15 |
| | RPCIC | 50.71 | 31.47 | 21.55 | 51.30 | 31.03 | 21.93 | _51.56_ | 31.84 | 24.20 | _53.34_ | 30.21 | 25.76 |
| | DCG | 47.38 | 30.27 | 24.67 | 46.29 | 29.57 | 25.00 | 50.04 | 30.74 | 26.24 | 49.60 | 30.35 | 26.00 |
| | BURG | 44.01 | 26.39 | 21.15 | 42.86 | 26.38 | 20.22 | 42.97 | 24.77 | 19.61 | 41.84 | 23.63 | 20.60 |
| | DIMVC-HIA | 49.82 | 34.57 | _28.62_ | 48.96 | 33.42 | _27.77_ | 47.34 | 28.90 | 23.67 | 49.95 | 33.14 | _27.42_ |
| | IDCD | **53.46** | _35.53_ | **29.23** | **53.32** | **34.90** | **29.14** | **54.55** | _35.18_ | **30.14** | **53.53** | **33.98** | **28.42** |

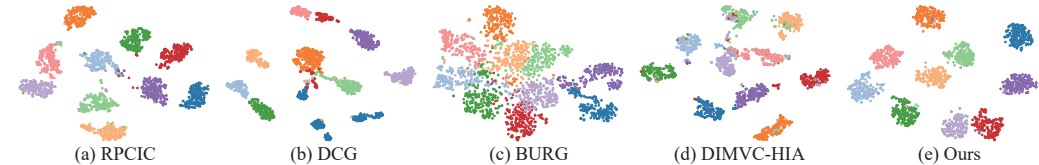

*Figure 2.* t-SNE visualization of the learned latent representations on the HandWritten dataset with a missing rate of 0.5.

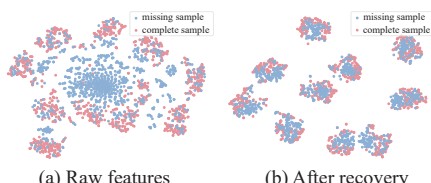

(a) Raw features    (b) After recovery

*Figure 3.* t-SNE visualization of the raw features and the recovered features on the HandWritten dataset with a missing rate of 0.5.

including DIMVC (Xu et al., 2022), DSIMVC (Tang & Liu, 2022), APADC (Xu et al., 2023), CPSPAN (Jin et al., 2023), DVIMC (Xu et al., 2024b), RPCIC (Yuan et al., 2024), DCG (Zhang et al., 2025), BURG (Jin et al., 2025), and DIMVC-HIA (Du et al., 2026). Among these methods, DIMVC, APADC and DVIMC are imputation-free approaches that directly operate on incomplete multi-view data, whereas the remaining methods rely on view imputation strategies to handle missing views. To quantitatively evaluate clustering performance, we adopt three widely used evaluation metrics: Clustering Accuracy (ACC), Normalized Mutual Information (NMI), and Adjusted Rand Index (ARI). Implementation details are provided in the Appendix B.2.

### 4.2. Experimental Results and Analysis

The quantitative clustering results under different missing rates are reported in Table 1. Notably, our approach achieves the best or second-best results in nearly all cases with respect to ACC, NMI, and ARI, indicating strong robustness to view incompleteness and stable clustering capability under varying degrees of missing data. On the CUB and HandWritten datasets, our method significantly outperforms all competing approaches under all missing rates. In particular, compared with strong baselines such as DCG and DVIMC, our model yields substantial performance gains, with improvements of up to approximately 6% in ACC, 2% in NMI, and 4% in ARI. These consistent improvements across multiple metrics suggest that the proposed method could maintain the robust performance even when a large portion of views is missing. Similar trends can be observed on more challenging datasets, including LandUse_21, CCV, and the large-scale Reuters dataset. Although these datasets exhibit higher intra-class variability and more complex data distributions, our method still maintains a clear advantage over most baselines under different missing rates. In particu-

lar, the performance gains remain stable as the missing rate increases from 0.1 to 0.7, demonstrating that the proposed approach is robust to view incompleteness and scales well to larger and more complex multi-view datasets. Furthermore, a clear performance difference can be observed between imputation-based and imputation-free methods. This observation confirms that missing views can introduce significant bias and information loss, which negatively affects clustering quality if not properly handled. The superior performance of our method further validates the effectiveness of incorporating principled view recovery mechanisms for incomplete multi-view clustering. We further conduct ablation studies under a missing rate of 0.1 by removing the MI constraint, replacing the diffusion module with an MLP, and replacing the proposed MWB with mean aggregation or product-of-experts (PoE). As shown in Tables 4 and 5, all variants result in performance degradation, demonstrating the effectiveness of MI-driven disentanglement, diffusion-based generation and Wasserstein-based latent aggregation. We further conduct parameter sensitivity analysis on the key hyperparameters, including $\alpha$, $\beta$, $\gamma$, the diffusion step $T$ and the mixture component number $K$. The results suggest that the proposed method is not overly sensitive to the choice of hyperparameters. Detailed experimental results are provided in the Appendix B.4.

### 4.3. Visualization Results and Analysis

To further illustrate the effectiveness of the proposed method, we visualize the learned latent representations $z$ on the HandWritten dataset with a missing rate of 0.5 using t-SNE (Maaten & Hinton, 2008), as shown in Figures 2 and 4. As can be observed, our method yields the most compact and well-separated clusters, where samples from the same class form clearly distinguishable groups with minimal overlap across different classes. In contrast, several competing methods exhibit inferior clustering structures. For example, BURG shows blurred cluster boundaries with noticeable inter-class mixing, while DIMVC-HIA suffers from a relatively large number of misclustered samples, indicating limited robustness to view incompleteness. These observations are consistent with the quantitative results reported in Table 1 and further demonstrate the superior discriminative capability of our method under missing-view scenarios.

To further assess the quality of the generated views, Figure 3

*Table 2.* Quantitative comparison of reconstruction performance on the CARL dataset under a missing rate of 0.1.

| View | Method | RMSE↓ | NMSE↓ | PSNR↑ |
|---|---|---|---|---|
| CLASSIC | DCG | 0.52 | 0.70 | 5.75 |
| | BURG | 0.20 | 0.11 | 13.85 |
| | IDCD | 0.19 | 0.09 | 14.58 |
| INFRARED | DCG | 0.61 | 0.85 | 4.34 |
| | BURG | 0.21 | 0.10 | 13.55 |
| | IDCD | 0.19 | 0.09 | 14.30 |
| THERMAL | DCG | 0.35 | 0.26 | 9.16 |
| | BURG | 0.16 | 0.06 | 15.70 |
| | IDCD | 0.14 | 0.04 | 17.09 |

presents the t-SNE visualizations of the raw features and the recovered features produced by our model under the same missing rate. As shown in Figure 3(a), due to view incompleteness, samples with missing views are clearly separated from those with complete views, leading to fragmented and distorted cluster structures. After applying our recovery mechanism, as illustrated in Figure 3(b), samples are reorganized into well-formed clusters according to their semantic categories, and missing samples become well aligned with complete ones. This indicates that the proposed model is able to effectively recover missing views while preserving the underlying class-specific semantic structure. Overall, these visual results confirm that our method enables robust and accurate incomplete multi-view clustering.

### 4.4. Disentanglement Results and Analysis

To evaluate the disentanglement between the shared latent representation $z$ and the view-specific latent variables $\{w^v\}$, we conduct both statistical dependence analysis and clustering-based evaluation under a missing rate of 0.1. Specifically, we employ the Hilbert–Schmidt Independence Criterion (HSIC) to measure the dependence between $z$ and each $w^v$, as well as between $z$ and the concatenated representation $w_{\text{concat}} = \text{Concat}([w^v])$. Lower HSIC values indicate weaker dependence and better disentanglement. In addition, clustering experiments are performed using $z$, each $w^v$, and $w_{\text{concat}}$, respectively. Since $z$ is expected to capture cross-view shared semantics while $w^v$ mainly encodes view-specific information, clustering performance based on $w^v$ should be significantly inferior to that achieved by $z$. The results summarized in Table 7 show that HSIC values remain consistently low across all datasets, indicating weak statistical dependence between shared and view-specific representations. Moreover, clustering performance based on $w^v$ and $w_{\text{concat}}$ is substantially lower than that achieved by $z$ in terms of ACC, NMI and ARI. These results demonstrate that $z$ effectively captures discriminative shared semantics, while $w^v$ mainly preserves modality-specific information, validating the disentanglement capability of the method.

### 4.5. Reconstruction Results and Analysis

To evaluate the reconstruction quality of the proposed method, we conduct quantitative experiments on the CARL dataset under a missing rate of 0.1. Three standard metrics are adopted, including Root Mean Square Error (RMSE), Normalized Mean Square Error (NMSE), and Peak Signal-to-Noise Ratio (PSNR), where lower RMSE/NMSE and higher PSNR indicate better reconstruction performance. We compare IDCD with representative generative-based baselines, including BURG and DCG, across three views (CLASSIC, INFRARED, and THERMAL). The results are summarized in Table 2. As shown, IDCD consistently outperforms BURG and DCG across all views and evaluation metrics, achieving the lowest RMSE and NMSE, as well as the highest PSNR. These results indicate that IDCD can effectively recover missing information and produce more accurate and stable reconstructions across heterogeneous modalities. In particular, the improvements are more pronounced on the THERMAL view, suggesting its stronger robustness under more challenging reconstruction scenarios. We further provide qualitative examples in Figure 8 to evaluate the visual quality of the reconstructed samples, where "Original" denotes the ground-truth images. As observed, IDCD generates more realistic and structurally consistent reconstructions across all views while preserving richer semantic details and modality-specific characteristics. Compared with BURG and DCG, the reconstructed results produced by IDCD exhibit clearer facial structures, sharper texture patterns, and higher visual fidelity, further validating its superior reconstruction capability under missing conditions.

## 5. Conclusion

We propose an information-guided disentangled framework with conditional diffusion for incomplete multi-view clustering. By decoupling shared semantics from view-specific factors and enforcing cross-view semantic consistency, the model achieves robust representation learning and reliable missing-view generation. Experimental results demonstrate consistent performance gains over state-of-the-art methods. Despite these promising results, several limitations remain. First, the current framework treats all modalities equally during feature aggregation, which may overlook the imbalance in information quality across views. Incorporating adaptive modality weighting strategies could further enhance representation learning and reconstruction performance. Second, the conditional diffusion module introduces additional computational overhead due to the iterative denoising process. Improving inference efficiency is therefore an important direction for future research. Finally, the generalization ability of the framework could be further strengthened by training on larger and more diverse multimodal datasets.

## Acknowledgements

This work was supported in part by the National Natural Science Foundation of China under Grant Nos. 62372279, 62322215 and 62532017, in part by the Natural Science Foundation of Shandong Province under Grant Nos. ZR2025QB62 and ZR2023MF119, in part by the Natural Science Foundation of Hunan Province under Grant No. 2026JJ30018 and in part by the High Performance Computing Center of Central South University.

## Impact Statement

This paper presents work whose goal is to advance the field of Machine Learning. There are many potential societal consequences of our work, none which we feel must be specifically highlighted here.

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

# A. Appendix

## A.1. Notation Details

*Table 3.* Mathematical Notation and Variable Definitions

| Symbol | Meaning | Space |
|---|---|---|
| $\boldsymbol{X}^v$ | Observed data in view $v$ | $\mathbb{R}^{n_v \times d_v}$ |
| $\boldsymbol{x}_i^v, \boldsymbol{x}^v$ | $i$-th instance in view $v$ | $\mathbb{R}^{d_v}$ |
| $\boldsymbol{g}^v$ | Binary indicator for observed samples | $\{0, 1\}^n$ |
| $\boldsymbol{h}_0^v$ | Encoded latent of view $v$ | $\mathbb{R}^{d_h}$ |
| $\boldsymbol{h}_t^v$ | Diffused latent representation at step $t$ | $\mathbb{R}^{d_h}$ |
| $\boldsymbol{z}^v$ | View-specific semantic latent | $\mathbb{R}^d$ |
| $\boldsymbol{w}^v$ | View-specific factor | $\mathbb{R}^{d_w}$ |
| $\boldsymbol{\mu}_{\boldsymbol{w}^v}$ | Mean of the variational distribution of $\boldsymbol{w}^v$ | $\mathbb{R}^{d_w}$ |
| $\boldsymbol{\sigma}_{\boldsymbol{w}^v}$ | Standard deviation of the variational distribution of $\boldsymbol{w}^v$ | $\mathbb{R}^{d_w}$ |
| $\boldsymbol{z}$ | Aggregated shared latent | $\mathbb{R}^d$ |
| $\boldsymbol{\sigma}_{\boldsymbol{z}^v}$ | Standard deviation of the variational distribution of $\boldsymbol{z}^v$ | $\mathbb{R}^d$ |
| $\boldsymbol{\mu}_{\boldsymbol{z}^v}$ | Mean of the variational distribution of $\boldsymbol{z}^v$ | $\mathbb{R}^d$ |
| $\{\boldsymbol{z}^v\}_{v=1}^V, \{\boldsymbol{z}^v\}$ | Set of view-invariant semantic latents | - |
| $\{\boldsymbol{w}^v\}_{v=1}^V, \{\boldsymbol{w}^v\}$ | Set of view-specific latent factors | - |
| $\{\boldsymbol{h}_{1:T}^v\}_{v=1}^V, \{\boldsymbol{h}_{1:T}^v\}$ | Diffusion trajectories for all views | - |
| $Z_m$ | One specific observed-view subset | $\subseteq \{1, \ldots, V\}$ |
| $\mathcal{Z}$ | Collection of all possible view subsets | $\mathcal{P}(\{1, \ldots, V\})$ |
| $c$ | Discrete cluster variable | $\{1, \ldots, K\}$ |
| $\tilde{\boldsymbol{\mu}}$ | Aggregated mean of $\{\boldsymbol{z}^v\}_{v \in Z_m}$ | $\mathbb{R}^d$ |
| $\tilde{\boldsymbol{\sigma}}$ | Aggregated standard deviation of $\{\boldsymbol{z}^v\}_{v \in Z_m}$ | $\mathbb{R}^d$ |
| $T$ | Number of diffusion steps | $\mathbb{N}_+$ |
| $\beta_t$ | Variance of the forward diffusion process at step $t$ | $\mathbb{R}_+$ |
| $\alpha_t$ | Noise retention coefficient at step $t$ | $\mathbb{R}_+$ |

## A.2. Bures-Wasserstein Barycenter Solution

**Theorem A.1** (Closed-form Wasserstein Barycenter for Isotropic Gaussians). *Let $\{q_\phi^j\}_{j=1}^M$ be isotropic Gaussian distributions $q_\phi^j = \mathcal{N}(\boldsymbol{\mu}_j, \sigma_j^2 \mathbf{I})$. The Wasserstein barycenter $q_{\mathrm{WB}} = \arg\min_q \sum_{j=1}^M \lambda_j \mathcal{W}_2^2(q_\phi^j, q), \sum_j \lambda_j = 1$, is also Gaussian $q_{WB} = \mathcal{N}(\tilde{\boldsymbol{\mu}}, \tilde{\boldsymbol{\sigma}}^2 \mathbf{I})$ with $\tilde{\boldsymbol{\mu}} = \sum_j \lambda_j \boldsymbol{\mu}_j, \tilde{\boldsymbol{\sigma}} = \sum_j \lambda_j \boldsymbol{\sigma}_j$ (Qiu et al., 2025).*

*Proof.* Assuming that each $q_{\phi_j} \sim \mathcal{N}(\boldsymbol{\mu}_j, \sigma_j^2 \mathbf{I})$ follows an isotropic Gaussian, the Bures-Wasserstein barycenter $q_{WB} \sim \mathcal{N}(\tilde{\boldsymbol{\mu}}, \tilde{\boldsymbol{\sigma}}^2 \mathbf{I})$ can be obtained by solving the following optimization problem:

$$q_{WB} = \arg\min_q \sum_{\boldsymbol{x}_j \in \boldsymbol{X}_m} \lambda_j \mathcal{W}_2^2(q_\phi^j, q).$$

Under the Gaussian assumption, the squared 2-Wasserstein distance between two isotropic Gaussians reduces to a simple sum of squared differences in means and standard deviations. Thus, the problem simplifies to:

$$\arg\min_{\tilde{\boldsymbol{\mu}}, \tilde{\boldsymbol{\sigma}}} \sum_{\boldsymbol{x}_j \in \boldsymbol{X}_m} \lambda_j \left[ (\tilde{\boldsymbol{\mu}} - \boldsymbol{\mu}_j)^2 + (\tilde{\boldsymbol{\sigma}} - \boldsymbol{\sigma}_j)^2 \right] := \arg\min_{\tilde{\boldsymbol{\mu}}, \tilde{\boldsymbol{\sigma}}} \mathcal{L}(\tilde{\boldsymbol{\mu}}, \tilde{\boldsymbol{\sigma}}).$$

To find the optimal barycenter parameters $\tilde{\boldsymbol{\mu}}$ and $\tilde{\boldsymbol{\sigma}}$, we compute the partial derivatives of the loss function $\mathcal{L}(\tilde{\boldsymbol{\mu}}, \tilde{\boldsymbol{\sigma}})$ with respect to each variable and set them to zero. This corresponds to finding the stationary point of a convex quadratic loss, which guarantees a global minimum:

$$\frac{\partial \mathcal{L}}{\partial \tilde{\boldsymbol{\mu}}} = 2 \sum_{\boldsymbol{x}_j \in \boldsymbol{X}_m} \lambda_j (\tilde{\boldsymbol{\mu}} - \boldsymbol{\mu}_j) = 0,$$

$$\frac{\partial \mathcal{L}}{\partial \tilde{\boldsymbol{\sigma}}} = 2 \sum_{\boldsymbol{x}_j \in \boldsymbol{X}_m} \lambda_j (\tilde{\boldsymbol{\sigma}} - \boldsymbol{\sigma}_j) = 0.$$

Solving these equations yields the optimal barycenter parameters as the weighted averages of the individual means and standard deviations:

$$\tilde{\boldsymbol{\mu}} = \sum_j \lambda_j \boldsymbol{\mu}_j, \quad \tilde{\boldsymbol{\sigma}} = \sum_j \lambda_j \boldsymbol{\sigma}_j.$$

These expressions provide the closed-form solution to the Bures-Wasserstein barycenter under the isotropic Gaussian assumption. $\square$

### A.3. Formulas on Mutual Information

In this section, we summarize some fundamental properties of mutual information that are utilized throughout this work.

- **(H1) Non-negativity:**

$$I(\mathbf{x}; \mathbf{y}) \geq 0, \quad I(\mathbf{x}; \mathbf{y} \mid \mathbf{z}) \geq 0, \tag{28}$$

  which states that both standard and conditional mutual information are always non-negative.

- **(H2) Chain rule:**

$$I(\mathbf{x}, \mathbf{y}; \mathbf{z}) = I(\mathbf{y}; \mathbf{z}) + I(\mathbf{x}; \mathbf{z} \mid \mathbf{y}), \tag{29}$$

  allowing the decomposition of mutual information between joint variables and a target variable.

- **(H3) Multivariate mutual information:**

$$I(\mathbf{x}; \mathbf{y}; \mathbf{z}) = I(\mathbf{x}; \mathbf{y}) - I(\mathbf{x}; \mathbf{y} \mid \mathbf{z}) = I(\mathbf{y}; \mathbf{z}) - I(\mathbf{y}; \mathbf{z} \mid \mathbf{x}), \tag{30}$$

  which quantifies the shared information among three variables and its reduction conditioned on one variable.

### A.4. Noise Predictor Implementation

The noise predictor is implemented using a U-Net backbone with hierarchical downsampling and upsampling blocks for multi-scale feature processing. To incorporate latent semantics into the denoising network, we adopt two complementary conditioning mechanisms.

First, both the shared semantic latent $\boldsymbol{z}$ and the view-specific latent $\boldsymbol{w}^v$ are injected through adaptive group normalization (AdaGN) layers via feature-wise affine modulation:

$$\mathrm{AdaGN}(\boldsymbol{h}, \boldsymbol{e}) = (1 + \boldsymbol{s}(\boldsymbol{e})) \cdot \mathrm{GroupNorm}(\boldsymbol{h}) + \boldsymbol{b}(\boldsymbol{e}), \tag{31}$$

where $\boldsymbol{e}$ denotes the conditioning embedding derived from $\boldsymbol{z}$ and $\boldsymbol{w}^v$, while $\boldsymbol{s}(\cdot)$ and $\boldsymbol{b}(\cdot)$ represent learnable scale and bias projections, respectively. This mechanism allows the conditioning latents to modulate intermediate feature statistics through feature-wise scaling and shifting.

Second, the shared semantic latent $\boldsymbol{z}$ is further incorporated through cross-attention layers to provide global semantic guidance during denoising:

$$\mathrm{CrossAttn}(\boldsymbol{H}, \boldsymbol{z}) = \mathrm{softmax}\left(\frac{Q(\boldsymbol{H})K(\boldsymbol{z})^\top}{\sqrt{d}}\right) V(\boldsymbol{z}), \tag{32}$$

where $Q(\cdot)$, $K(\cdot)$, and $V(\cdot)$ denote the query, key, and value projections, respectively. Here, intermediate spatial representations attend to the shared semantic latent $\boldsymbol{z}$ as global semantic context, thereby enhancing cross-view semantic consistency during generation.

## A.5. Derivation of the Evidence Lower Bound for Eq. 19

$$\log p(\{\boldsymbol{h}_0^v\}) = \log \int p(\{\boldsymbol{h}_{0:T}^v\}, \boldsymbol{z}, \{\boldsymbol{w}^v\}, \boldsymbol{c}) d\{\boldsymbol{h}_{1:T}^v\} d\boldsymbol{z} d\{\boldsymbol{w}^v\} d\boldsymbol{c}$$

$$= \log \int \frac{p(\{\boldsymbol{h}_{0:T}^v\}, \boldsymbol{z}, \{\boldsymbol{w}^v\}, \boldsymbol{c}) q(\boldsymbol{z} \mid \{\boldsymbol{h}_0^v\}) q(\boldsymbol{c} \mid \boldsymbol{z}) \prod_{v=1}^{V} q(\boldsymbol{h}_{1:T}^v \mid \boldsymbol{h}_0^v) q(\boldsymbol{w}^v \mid \boldsymbol{h}_0^v)}{q(\boldsymbol{z} \mid \{\boldsymbol{h}_0^v\}) q(\boldsymbol{c} \mid \boldsymbol{z}) \prod_{v=1}^{V} q(\boldsymbol{h}_{1:T}^v \mid \boldsymbol{h}_0^v) q(\boldsymbol{w}^v \mid \boldsymbol{h}_0^v)} d\{\boldsymbol{h}_{1:T}^v\} d\boldsymbol{z} d\{\boldsymbol{w}^v\} d\boldsymbol{c}$$

$$\geq \mathbb{E}_{q(\boldsymbol{z}|\{\boldsymbol{h}_0^v\}) q(\boldsymbol{c}|\boldsymbol{z}) \prod_{v=1}^{V} q(\boldsymbol{h}_{1:T}^v|\boldsymbol{h}_0^v) q(\boldsymbol{w}^v|\boldsymbol{h}_0^v)} \left[ \log \frac{p(\boldsymbol{c}) p(\boldsymbol{z} \mid \boldsymbol{c}) \prod_{v=1}^{V} p(\boldsymbol{w}^v) p(\boldsymbol{h}_T^v) \prod_{t=1}^{T} p(\boldsymbol{h}_{t-1}^v \mid \boldsymbol{h}_t^v, \boldsymbol{z}, \boldsymbol{w}^v)}{q(\boldsymbol{z} \mid \{\boldsymbol{h}_0^v\}) q(\boldsymbol{c} \mid \boldsymbol{z}) \prod_{v=1}^{V} q(\boldsymbol{h}_{1:T}^v \mid \boldsymbol{h}_0^v) q(\boldsymbol{w}^v \mid \boldsymbol{h}_0^v)} \right]$$

$$= \mathbb{E}_{q(\boldsymbol{z}|\{\boldsymbol{h}_0^v\}) q(\boldsymbol{c}|\boldsymbol{z}) \prod_{v=1}^{V} q(\boldsymbol{h}_{1:T}^v|\boldsymbol{h}_0^v) q(\boldsymbol{w}^v|\boldsymbol{h}_0^v)} \left[ \log \frac{p(\boldsymbol{c})}{q(\boldsymbol{c} \mid \boldsymbol{z})} + \log \frac{p(\boldsymbol{z} \mid \boldsymbol{c})}{q(\boldsymbol{z} \mid \{\boldsymbol{h}_0^v\})} + \sum_{v=1}^{V} \log \frac{p(\boldsymbol{w}^v)}{q(\boldsymbol{w}^v \mid \boldsymbol{h}_0^v)} \right]$$

$$+ \mathbb{E}_{q(\boldsymbol{z}|\{\boldsymbol{h}_0^v\}) q(\boldsymbol{c}|\boldsymbol{z}) \prod_{v=1}^{V} q(\boldsymbol{h}_{1:T}^v|\boldsymbol{h}_0^v) q(\boldsymbol{w}^v|\boldsymbol{h}_0^v)} \left[ \sum_{v=1}^{V} \log \frac{p(\boldsymbol{h}_T^v) \prod_{t=1}^{T} p(\boldsymbol{h}_{t-1}^v \mid \boldsymbol{h}_t^v, \boldsymbol{z}, \boldsymbol{w}^v)}{\prod_{t=1}^{T} q(\boldsymbol{h}_t^v \mid \boldsymbol{h}_{t-1}^v)} \right]$$

$$= \mathbb{E}_{q(\boldsymbol{z}|\{\boldsymbol{h}_0^v\}) q(\boldsymbol{c}|\boldsymbol{z}) \prod_{v=1}^{V} q(\boldsymbol{h}_{1:T}^v|\boldsymbol{h}_0^v) q(\boldsymbol{w}^v|\boldsymbol{h}_0^v)} \left[ \log \frac{p(\boldsymbol{c})}{q(\boldsymbol{c} \mid \boldsymbol{z})} + \log \frac{p(\boldsymbol{z} \mid \boldsymbol{c})}{q(\boldsymbol{z} \mid \{\boldsymbol{h}_0^v\})} + \sum_{v=1}^{V} \log \frac{p(\boldsymbol{w}^v)}{q(\boldsymbol{w}^v \mid \boldsymbol{h}_0^v)} \right]$$

$$+ \mathbb{E}_{q(\boldsymbol{z}|\{\boldsymbol{h}_0^v\}) q(\boldsymbol{c}|\boldsymbol{z}) \prod_{v=1}^{V} q(\boldsymbol{h}_{1:T}^v|\boldsymbol{h}_0^v) q(\boldsymbol{w}^v|\boldsymbol{h}_0^v)} \left[ \sum_{v=1}^{V} \log \frac{p(\boldsymbol{h}_T^v) p(\boldsymbol{h}_0^v \mid \boldsymbol{h}_1^v, \boldsymbol{z}, \boldsymbol{w}^v) \prod_{t=2}^{T} p(\boldsymbol{h}_{t-1}^v \mid \boldsymbol{h}_t^v, \boldsymbol{z}, \boldsymbol{w}^v)}{q(\boldsymbol{h}_1^v \mid \boldsymbol{h}_0^v) \prod_{t=2}^{T} q(\boldsymbol{h}_t^v \mid \boldsymbol{h}_{t-1}^v, \boldsymbol{h}_0^v)} \right]$$

$$= \mathbb{E}_{q(\boldsymbol{z}|\{\boldsymbol{h}_0^v\}) q(\boldsymbol{c}|\boldsymbol{z}) \prod_{v=1}^{V} q(\boldsymbol{h}_{1:T}^v|\boldsymbol{h}_0^v) q(\boldsymbol{w}^v|\boldsymbol{h}_0^v)} \left[ \log \frac{p(\boldsymbol{c})}{q(\boldsymbol{c} \mid \boldsymbol{z})} + \log \frac{p(\boldsymbol{z} \mid \boldsymbol{c})}{q(\boldsymbol{z} \mid \{\boldsymbol{h}_0^v\})} + \sum_{v=1}^{V} \log \frac{p(\boldsymbol{w}^v)}{q(\boldsymbol{w}^v \mid \boldsymbol{h}_0^v)} \right]$$

$$+ \mathbb{E}_{q(\boldsymbol{z}|\{\boldsymbol{h}_0^v\}) q(\boldsymbol{c}|\boldsymbol{z}) \prod_{v=1}^{V} q(\boldsymbol{h}_{1:T}^v|\boldsymbol{h}_0^v) q(\boldsymbol{w}^v|\boldsymbol{h}_0^v)} \left[ \sum_{v=1}^{V} \log \frac{p(\boldsymbol{h}_T^v) p(\boldsymbol{h}_0^v \mid \boldsymbol{h}_1^v, \boldsymbol{z}, \boldsymbol{w}^v)}{q(\boldsymbol{h}_1^v \mid \boldsymbol{h}_0^v)} + \sum_{v=1}^{V} \log \frac{\prod_{t=2}^{T} p(\boldsymbol{h}_{t-1}^v \mid \boldsymbol{h}_t^v, \boldsymbol{z}, \boldsymbol{w}^v)}{\prod_{t=2}^{T} \frac{q(\boldsymbol{h}_{t-1}^v|\boldsymbol{h}_t^v, \boldsymbol{h}_0^v) q(\boldsymbol{h}_t^v|\boldsymbol{h}_0^v)}{q(\boldsymbol{h}_{t-1}^v|\boldsymbol{h}_0^v)}} \right]$$

$$= \mathbb{E}_{q(\boldsymbol{z}|\{\boldsymbol{h}_0^v\}) q(\boldsymbol{c}|\boldsymbol{z}) \prod_{v=1}^{V} q(\boldsymbol{h}_{1:T}^v|\boldsymbol{h}_0^v) q(\boldsymbol{w}^v|\boldsymbol{h}_0^v)} \left[ \log \frac{p(\boldsymbol{c})}{q(\boldsymbol{c} \mid \boldsymbol{z})} + \log \frac{p(\boldsymbol{z} \mid \boldsymbol{c})}{q(\boldsymbol{z} \mid \{\boldsymbol{h}_0^v\})} + \sum_{v=1}^{V} \log \frac{p(\boldsymbol{w}^v)}{q(\boldsymbol{w}^v \mid \boldsymbol{h}_0^v)} \right]$$

$$+ \mathbb{E}_{q(\boldsymbol{z}|\{\boldsymbol{h}_0^v\}) q(\boldsymbol{c}|\boldsymbol{z}) \prod_{v=1}^{V} q(\boldsymbol{h}_{1:T}^v|\boldsymbol{h}_0^v) q(\boldsymbol{w}^v|\boldsymbol{h}_0^v)} \left[ \sum_{v=1}^{V} \log \frac{p(\boldsymbol{h}_T^v) p(\boldsymbol{h}_0^v \mid \boldsymbol{h}_1^v, \boldsymbol{z}, \boldsymbol{w}^v) \cancel{q(\boldsymbol{h}_1^v|\boldsymbol{h}_0^v)}}{\cancel{q(\boldsymbol{h}_1^v|\boldsymbol{h}_0^v)} q(\boldsymbol{h}_T^v \mid \boldsymbol{h}_0^v)} + \sum_{v=1}^{V} \log \frac{\prod_{t=2}^{T} p(\boldsymbol{h}_{t-1}^v \mid \boldsymbol{h}_t^v, \boldsymbol{z}, \boldsymbol{w}^v)}{\prod_{t=2}^{T} q(\boldsymbol{h}_{t-1}^v \mid \boldsymbol{h}_t^v, \boldsymbol{h}_0^v)} \right]$$

$$= \sum_{v=1}^{V} \mathbb{E}_{q(\boldsymbol{h}_1^v|\boldsymbol{h}_0^v)} \mathbb{E}_{q(\boldsymbol{z}|\{\boldsymbol{h}_0^v\}) q(\boldsymbol{w}^v|\boldsymbol{h}_0^v)} [\log p(\boldsymbol{h}_0^v \mid \boldsymbol{h}_1^v, \boldsymbol{z}, \boldsymbol{w}^v)] - \sum_{v=1}^{V} KL\left(q(\boldsymbol{h}_T^v \mid \boldsymbol{h}_0^v) \| p(\boldsymbol{h}_T^v)\right)$$

$$- \sum_{v=1}^{V} \sum_{t=2}^{T} \mathbb{E}_{q(\boldsymbol{h}_t^v|\boldsymbol{h}_0^v)} \mathbb{E}_{q(\boldsymbol{z}|\{\boldsymbol{h}_0^v\}) q(\boldsymbol{w}^v|\boldsymbol{h}_0^v)} \left[ KL\left(q(\boldsymbol{h}_{t-1}^v \mid \boldsymbol{h}_t^v, \boldsymbol{h}_0^v) \| p(\boldsymbol{h}_{t-1}^v \mid \boldsymbol{h}_t^v, \boldsymbol{z}, \boldsymbol{w}^v)\right) \right]$$

$$- \mathbb{E}_{q(\boldsymbol{z}|\{\boldsymbol{h}_0^v\})} \left[ KL\left(q(\boldsymbol{c} \mid \boldsymbol{z}) \| p(\boldsymbol{c})\right) \right] - \mathbb{E}_{q(\boldsymbol{c}|\boldsymbol{z})} \left[ KL\left(q(\boldsymbol{z} \mid \{\boldsymbol{h}_0^v\}) \| p(\boldsymbol{z} \mid \boldsymbol{c})\right) \right] - \sum_{v=1}^{V} KL\left(q(\boldsymbol{w}^v \mid \boldsymbol{h}_0^v) \| p(\boldsymbol{w}^v)\right).$$

## A.6. Definition of Latent Semantic Properties

**Definition A.2** (Latent Factor Disentanglement). Let $\boldsymbol{z}$ denote the shared latent variable and $\{\boldsymbol{w}^v\}_{v=1}^{V}$ denote the set of view-specific latent variables. The latent representation is said to satisfy disentanglement if the generative factors encoded by $\boldsymbol{z}$ and $\{\boldsymbol{w}^v\}$ are mutually exclusive, such that no factor of variation is redundantly represented across the shared and view-specific latent subspaces.

**Definition A.3** (Latent Semantic Sufficiency). Let $\{\boldsymbol{h}_0^v\}_{v=1}^{V}$ denote the observed multi-view features, $\boldsymbol{z}$ is the shared latent variable and $\{\boldsymbol{w}^v\}_{v=1}^{V}$ are the view-specific latent variables. The latent representation $(\boldsymbol{z}, \{\boldsymbol{w}^v\})$ is said to be semantically sufficient if it preserves all information from $\{\boldsymbol{h}_0^v\}$ that is relevant for reconstructing or generating the observations, i.e., $I(\{\boldsymbol{h}_0^v\}; \boldsymbol{z}, \{\boldsymbol{w}^v\})$ is maximized.

## A.7. Structural Information Bound

*Proof.* Starting from the mutual information objective, we have:

$$I(\{\boldsymbol{h}_0^v\}; \boldsymbol{z}, \{\boldsymbol{w}^v\}) - I(\boldsymbol{z}; \{\boldsymbol{w}^v\}).$$

To derive a tractable expression for $I(\boldsymbol{z}; \{\boldsymbol{w}^v\})$, we apply Eq. (30) to the three random variables $\boldsymbol{z}$, $\{\boldsymbol{w}^v\}$, and $\{\boldsymbol{h}_0^v\}$. Specifically, setting $X = \{\boldsymbol{h}_0^v\}$, $Y = \boldsymbol{z}$, and $Z = \{\boldsymbol{w}^v\}$, we obtain

$$I(\boldsymbol{z}; \{\boldsymbol{w}^v\}) = I(\boldsymbol{z}; \{\boldsymbol{h}_0^v\}) - I(\boldsymbol{z}; \{\boldsymbol{h}_0^v\} \mid \{\boldsymbol{w}^v\}) + I(\boldsymbol{z}; \{\boldsymbol{w}^v\} \mid \{\boldsymbol{h}_0^v\}).$$

Due to the structural assumption on $q$, we have $q(\boldsymbol{z} \mid \{\boldsymbol{h}_0^v\}) = q(\boldsymbol{z} \mid \{\boldsymbol{h}_0^v\}, \{\boldsymbol{w}^v\})$, which implies the conditional independence $\boldsymbol{z} \perp \{\boldsymbol{w}^v\} \mid \{\boldsymbol{h}_0^v\}$. Thus, the last term in the above equation disappears:

$$I(\boldsymbol{z}; \{\boldsymbol{w}^v\} \mid \{\boldsymbol{h}_0^v\}) = H(\boldsymbol{z} \mid \{\boldsymbol{h}_0^v\}) - H(\boldsymbol{z} \mid \{\boldsymbol{h}_0^v\}, \{\boldsymbol{w}^v\}) = H(\boldsymbol{z} \mid \{\boldsymbol{h}_0^v\}) - H(\boldsymbol{z} \mid \{\boldsymbol{h}_0^v\}) = 0.$$

We can rewrite

$$I(\boldsymbol{z}; \{\boldsymbol{w}^v\}) = I(\boldsymbol{z}; \{\boldsymbol{h}_0^v\}) - I(\boldsymbol{z}; \{\boldsymbol{h}_0^v\} \mid \{\boldsymbol{w}^v\}) = I(\boldsymbol{z}; \{\boldsymbol{h}_0^v\}) + I(\{\boldsymbol{w}^v\}; \{\boldsymbol{h}_0^v\}) - I(\{\boldsymbol{h}_0^v\}; \boldsymbol{z}, \{\boldsymbol{w}^v\}).$$

Substituting this into $\mathcal{L}_{\mathrm{dis}}$ and collecting terms yields:

$$\mathcal{L}_{\mathrm{dis}} = 2I(\{\boldsymbol{h}_0^v\}; \boldsymbol{z}, \{\boldsymbol{w}^v\}) - I(\{\boldsymbol{h}_0^v\}; \boldsymbol{z}) - I(\{\boldsymbol{h}_0^v\}; \{\boldsymbol{w}^v\}).$$

Since mutual information satisfies Eqs. (28) and (29), we have

$$I(\{\boldsymbol{h}_0^v\}; \boldsymbol{z}, \boldsymbol{c}) = I(\{\boldsymbol{h}_0^v\}; \boldsymbol{z}) + I(\{\boldsymbol{h}_0^v\}; \boldsymbol{c} \mid \boldsymbol{z}) \geq I(\{\boldsymbol{h}_0^v\}; \boldsymbol{z}).$$

Hence,

$$I(\{\boldsymbol{h}_0^v\}; \boldsymbol{z}) \leq I(\{\boldsymbol{h}_0^v\}; \boldsymbol{z}, \boldsymbol{c}).$$

Substituting this inequality into the expression of $\mathcal{L}_{\mathrm{dis}}$, we obtain

$$\mathcal{L}_{\mathrm{dis}} \geq 2I(\{\boldsymbol{h}_0^v\}; \boldsymbol{z}, \{\boldsymbol{w}^v\}) - I(\{\boldsymbol{h}_0^v\}; \boldsymbol{z}, \boldsymbol{c}) - I(\{\boldsymbol{h}_0^v\}; \{\boldsymbol{w}^v\}).$$

According to the definition of mutual information, we have

$$\begin{aligned}
\mathcal{L}_{\mathrm{dis}} \geq \ & 2\mathbb{E}_{q(\{\boldsymbol{h}_0^v\})}\mathbb{E}_{q(\boldsymbol{z}, \{\boldsymbol{w}^v\} \mid \{\boldsymbol{h}_0^v\})} \log \frac{q(\{\boldsymbol{h}_0^v\} \mid \boldsymbol{z}, \{\boldsymbol{w}^v\})}{q(\{\boldsymbol{h}_0^v\})} \\
& - \mathbb{E}_{q(\{\boldsymbol{h}_0^v\})}\mathbb{E}_{q(\boldsymbol{z} \mid \{\boldsymbol{h}_0^v\})q(\boldsymbol{c} \mid \boldsymbol{z})} \log \frac{q(\boldsymbol{z}, \boldsymbol{c} \mid \{\boldsymbol{h}_0^v\})}{q(\boldsymbol{z}, \boldsymbol{c})} \\
& - \mathbb{E}_{q(\{\boldsymbol{h}_0^v\})}\mathbb{E}_{q(\{\boldsymbol{w}^v\} \mid \{\boldsymbol{h}_0^v\})} \log \frac{q(\{\boldsymbol{w}^v\} \mid \{\boldsymbol{h}_0^v\})}{q(\{\boldsymbol{w}^v\})}.
\end{aligned}$$

By introducing the generative model $p(\{\boldsymbol{h}_0^v\} \mid \boldsymbol{z}, \{\boldsymbol{w}^v\})$ and the hierarchical priors $p(\boldsymbol{z} \mid \boldsymbol{c})$ and $p(\boldsymbol{w}^v)$, the above expression can be decomposed as

$$\begin{aligned}
\mathcal{L}_{\mathrm{dis}} \geq \ & 2\mathbb{E}_{q(\{\boldsymbol{h}_0^v\})}\mathbb{E}_{q(\boldsymbol{z}, \{\boldsymbol{w}^v\} \mid \{\boldsymbol{h}_0^v\})} \log p(\{\boldsymbol{h}_0^v\} \mid \boldsymbol{z}, \{\boldsymbol{w}^v\}) \\
& + 2\mathbb{E}_{q(\boldsymbol{z}, \{\boldsymbol{w}^v\})} \Big[ KL\big( q(\{\boldsymbol{h}_0^v\} \mid \boldsymbol{z}, \{\boldsymbol{w}^v\}) \,\|\, p(\{\boldsymbol{h}_0^v\} \mid \boldsymbol{z}, \{\boldsymbol{w}^v\}) \big) \Big] \\
& - \mathbb{E}_{q(\{\boldsymbol{h}_0^v\})}\mathbb{E}_{q(\boldsymbol{c} \mid \boldsymbol{z})} \Big[ KL\big( q(\boldsymbol{z} \mid \{\boldsymbol{h}_0^v\}) \,\|\, p(\boldsymbol{z} \mid \boldsymbol{c}) \big) \Big] \\
& + \mathbb{E}_{q(\boldsymbol{c} \mid \boldsymbol{z})} \Big[ KL\big( q(\boldsymbol{z}) \,\|\, p(\boldsymbol{z} \mid \boldsymbol{c}) \big) \Big] \\
& - \mathbb{E}_{q(\{\boldsymbol{h}_0^v\})} \Big[ KL\big( q(\{\boldsymbol{w}^v\} \mid \{\boldsymbol{h}_0^v\}) \,\|\, p(\{\boldsymbol{w}^v\}) \big) \Big] \\
& + KL\big( q(\{\boldsymbol{w}^v\}) \,\|\, p(\{\boldsymbol{w}^v\}) \big).
\end{aligned}$$

Since KL divergences are non-negative, discarding the non-negative terms yields the following lower bound:

$$\mathcal{L}_{\text{dis}} \geq 2\mathbb{E}_{q(\{\boldsymbol{h}_0^v\})}\mathbb{E}_{q(\boldsymbol{z},\{\boldsymbol{w}^v\}|\{\boldsymbol{h}_0^v\})} \log p(\{\boldsymbol{h}_0^v\} \mid \boldsymbol{z}, \{\boldsymbol{w}^v\})$$
$$- \mathbb{E}_{q(\{\boldsymbol{h}_0^v\})}\mathbb{E}_{q(\boldsymbol{c}|\boldsymbol{z})}\Big[KL\big(q(\boldsymbol{z} \mid \{\boldsymbol{h}_0^v\}) \,\|\, p(\boldsymbol{z} \mid \boldsymbol{c})\big)\Big]$$
$$- \mathbb{E}_{q(\{\boldsymbol{h}_0^v\})}\Big[KL\big(q(\{\boldsymbol{w}^v\} \mid \{\boldsymbol{h}_0^v\}) \,\|\, p(\{\boldsymbol{w}^v\})\big)\Big],$$

where the three terms respectively denote a reconstruction likelihood for semantic sufficiency, a clustering-based regularizer for shared-space consistency and view-specific KL penalties serving as information bottlenecks for factor disentanglement. This establishes the claimed variational lower bound for the mutual information objective. $\square$

## A.8. Contrastive Alignment Lower Bound

*Proof.* For two random variables $\boldsymbol{z}^v$ and $\boldsymbol{z}^l$, the mutual information is defined as

$$I(\boldsymbol{z}^v; \boldsymbol{z}^l) = \mathbb{E}_{p(\boldsymbol{z}^v, \boldsymbol{z}^l)}\left[\log \frac{p(\boldsymbol{z}^v, \boldsymbol{z}^l)}{p(\boldsymbol{z}^v)p(\boldsymbol{z}^l)}\right] = \mathbb{E}_{p(\boldsymbol{z}^v, \boldsymbol{z}^l)}\left[\log \frac{p(\boldsymbol{z}^l \mid \boldsymbol{z}^v)}{p(\boldsymbol{z}^l)}\right]. \tag{33}$$

Let $f_\theta(\boldsymbol{z}^v, \boldsymbol{z}^l)$ be a critic function that estimates the density ratio $\frac{p(\boldsymbol{z}^l|\boldsymbol{z}^v)}{p(\boldsymbol{z}^l)}$. Following the variational formulation of mutual information, for any positive function $f_\theta$, we have

$$I(\boldsymbol{z}^v; \boldsymbol{z}^l) \geq \mathbb{E}_{p(\boldsymbol{z}^v, \boldsymbol{z}^l)}\left[\log \frac{f_\theta(\boldsymbol{z}^v, \boldsymbol{z}^l)}{\mathbb{E}_{p(\tilde{\boldsymbol{z}}^l)}[f_\theta(\boldsymbol{z}^v, \tilde{\boldsymbol{z}}^l)]}\right], \tag{34}$$

where $\tilde{\boldsymbol{z}}^l \sim p(\boldsymbol{z}^l)$ are samples drawn independently from the marginal. This inequality follows from Jensen's inequality and becomes tight when $f_\theta(\boldsymbol{z}^v, \boldsymbol{z}^l) \propto \frac{p(\boldsymbol{z}^l|\boldsymbol{z}^v)}{p(\boldsymbol{z}^l)}$.

In practice, the marginal expectation in the denominator is approximated using $N$ samples in a minibatch. For each anchor $\boldsymbol{z}_i^v$, the positive sample is $\boldsymbol{z}_i^l$ from the same instance, and $\{\boldsymbol{z}_j^l\}_{j=1}^N$ serve as Monte Carlo samples from the marginal $p(\boldsymbol{z}^l)$. Choosing the critic in exponential form,

$$f_\theta(\boldsymbol{z}^v, \boldsymbol{z}^l) = \exp\left(\frac{s(\boldsymbol{z}^v, \boldsymbol{z}^l)}{\tau}\right), \tag{35}$$

where $s(\cdot, \cdot)$ is a similarity score and $\tau > 0$ is a temperature, Eq. (34) yields the InfoNCE estimator:

$$I(\boldsymbol{z}^v; \boldsymbol{z}^l) \geq \mathbb{E}_i\left[\log \frac{\exp\big(s(\boldsymbol{z}_i^v, \boldsymbol{z}_i^l)/\tau\big)}{\sum_{j=1}^N \exp\big(s(\boldsymbol{z}_i^v, \boldsymbol{z}_j^l)/\tau\big)}\right]. \tag{36}$$

Finally, summing the bound over all ordered view pairs $(v, l)$ with $v \neq l$ gives

$$\sum_{v=1}^V \sum_{l \neq v} I(\boldsymbol{z}^v; \boldsymbol{z}^l) \geq \sum_{v=1}^V \sum_{l \neq v} \mathbb{E}_i\left[\log \frac{\exp\big(s(\boldsymbol{z}_i^v, \boldsymbol{z}_i^l)/\tau\big)}{\sum_{j=1}^N \exp\big(s(\boldsymbol{z}_i^v, \boldsymbol{z}_j^l)/\tau\big)}\right], \tag{37}$$

which proves the claimed contrastive lower bound. $\square$

## A.9. Upper Bound on Expected Gaussian Log-Likelihood

**Lemma A.4.** *Let $\boldsymbol{x} \in \mathbb{R}^J$ be a random variable with density $p(\boldsymbol{x})$. Consider a Gaussian density $q(\boldsymbol{x}) = \mathcal{N}(\boldsymbol{x}; \boldsymbol{\mu}, \mathbf{I})$, with mean $\boldsymbol{\mu}$ and identity covariance. Then the expected log-likelihood of $q$ under $p$ satisfies*

$$\mathbb{E}_{p(\boldsymbol{x})}[\log q(\boldsymbol{x})] \leq -\frac{J}{2}\log(2\pi) < 0. \tag{38}$$

This lemma shows that the expected log-likelihood under an isotropic Gaussian admits a dimension-dependent constant upper bound.

*Proof.* The probability density function of a multivariate Gaussian with mean $\boldsymbol{\mu} \in \mathbb{R}^J$ and covariance $\mathbf{I}$ is given by:

$$q(\boldsymbol{x}) = \frac{1}{(2\pi)^{J/2}} \exp\left(-\frac{1}{2}\|\boldsymbol{x} - \boldsymbol{\mu}\|^2\right)$$

which implies the log-density is:

$$\log q(\boldsymbol{x}) = -\frac{J}{2}\log(2\pi) - \frac{1}{2}\|\boldsymbol{x} - \boldsymbol{\mu}\|^2$$

Taking expectation with respect to $p(\boldsymbol{x})$, we have:

$$\mathbb{E}_{p(\boldsymbol{x})}[\log q(\boldsymbol{x})] = -\frac{J}{2}\log(2\pi) - \frac{1}{2}\mathbb{E}_{p(\boldsymbol{x})}[\|\boldsymbol{x} - \boldsymbol{\mu}\|^2]$$

since $p(\boldsymbol{x})$ has finite second moments, the expectation $\mathbb{E}_{p(\boldsymbol{x})}[\|\boldsymbol{x} - \boldsymbol{\mu}\|^2]$ is finite and non-negative. Therefore,

$$\mathbb{E}_{p(\boldsymbol{x})}[\log q(\boldsymbol{x})] \leq -\frac{J}{2}\log(2\pi) < 0$$

$\square$

### A.10. Upper Bound Property of the Contrastive Log-Softmax Term

**Theorem A.5** (Upper Bound Property of the Contrastive Log-Softmax Term). *Consider the standard contrastive learning setting with an anchor $\boldsymbol{z}_i^v$, its positive sample $\boldsymbol{z}_i^l$, and a set of $N-1$ negative samples $\{\boldsymbol{z}_j^l\}_{j \neq i}$. Let $s(\cdot, \cdot)$ be a similarity function and $\tau > 0$ a temperature parameter. Define the InfoNCE probability of correctly identifying the positive pair as*

$$p_i = \frac{\exp\big(s(\boldsymbol{z}_i^v, \boldsymbol{z}_i^l)/\tau\big)}{\sum_{j=1}^{N} \exp\big(s(\boldsymbol{z}_i^v, \boldsymbol{z}_j^l)/\tau\big)}.$$

*Then the per-sample contrastive log-likelihood term satisfies*

$$\log p_i \leq 0,$$

*and consequently the InfoNCE loss*

$$\mathcal{L}_{\text{InfoNCE}} = -\mathbb{E}_i[\log p_i] \geq 0$$

.

*Proof.* In contrastive learning, the denominator contains the positive pair and all negatives:

$$\sum_{j=1}^{N} \exp(s(\boldsymbol{z}_i^v, \boldsymbol{z}_j^l)/\tau) \geq \exp(s(\boldsymbol{z}_i^v, \boldsymbol{z}_i^l)/\tau).$$

Thus

$$0 < p_i \leq 1.$$

Since the logarithm is monotonic,

$$\log p_i \leq 0.$$

Taking expectation over samples gives

$$\mathbb{E}_i[\log p_i] \leq 0 \quad \Rightarrow \quad \mathcal{L}_{\text{InfoNCE}} = -\mathbb{E}_i[\log p_i] \geq 0.$$

$\square$

## A.11. Algorithm

---

**Algorithm 1** IDCD

---

**Require:** Multi-view data $\{\boldsymbol{x}^v\}_{v=1}^V$, indicator matrices $\{\boldsymbol{g}^v\}_{v=1}^V$, number of clusters $K$, diffusion steps $T$, missing rate $\eta$, noise schedule $\alpha_t$, pretraining learning rate $\rho_1$, training learning rate $\rho_2$

**Ensure:** Cluster assignments $\boldsymbol{y}$

1: **Stage 1: View-specific encoder pretraining**
2: **for** $v = 1$ to $V$ **do**
3:      Pretrain the encoder $E^v$ on $\boldsymbol{X}^v$ using the reconstruction loss of an autoencoder with learning rate $\rho_1$.
4: **end for**
5: **Stage 2: Pretraining on complete paired data**
6: Initialize trainable parameters $\{\mathcal{H}_z^v, \mathcal{H}_w^v\}_{v=1}^V$, $\epsilon_\theta$, and $\{\boldsymbol{\pi}_k, \boldsymbol{\mu}_k, \boldsymbol{\sigma}_k\}_{k=1}^K$ while keeping the pretrained encoders $\{E^v\}_{v=1}^V$ fixed
7: Compute $\boldsymbol{g}^{\text{all}} \leftarrow \bigwedge_{v=1}^V \boldsymbol{g}^v$     // fully observed samples
8: **for** each iteration **do**
9:      **if** $\boldsymbol{g}^{\text{all}} = 1$ **then**
10:          **for** $v = 1$ to $V$ **do**
11:              Compute $\boldsymbol{h}_0^v$ by applying $E^v$
12:              Compute the parameters of the variational distributions: $[\boldsymbol{\mu}_{w^v}, \boldsymbol{\sigma}_{w^v}^2] = \mathcal{H}_w^v(h_0^v)$ and $[\boldsymbol{\mu}_{z^v}, \boldsymbol{\sigma}_{z^v}^2] = \mathcal{H}_z^v(h_0^v)$
13:              Compute view-specific factors $\boldsymbol{w}^v$ by Eq. (7)
14:              Define the forward diffusion process over latent variables $\boldsymbol{h}_t^v$ as in Eq. (14)
15:          **end for**
16:          Compute the shared latent representation $\boldsymbol{z}$ using the mixture of Wasserstein barycenter (Eq. (11))
17:          Infer the cluster assignment $\boldsymbol{c}$ from the shared latent $\boldsymbol{z}$ according to Eq. (12)
18:          Train the noise prediction network $\epsilon_\theta(\cdot)$ to estimate the diffusion noise
19:          Compute the ELBO objective by Eq. (19)
20:          Compute mutual information $\mathcal{L}_{\text{dis}}$ by Eq. (24) for semantic sufficiency and disentanglement
21:          Compute mutual information $\mathcal{L}_{\text{InfoNCE}}$ by Eq. (26) for cross-view semantic alignment
22:          Optimize the overall training loss by Eq. (27)
23:      **end if**
24: **end for**
25: **Stage 3: Train on all data**
26: Same as Stage 2, but use available samples per view indicated by $\boldsymbol{g}^v = 1$
27: **Stage 4: Missing view imputation**
28: Encode $\boldsymbol{h}_0^v = E^v(x^v)$, and sample view-specific $\boldsymbol{w}^v$ and shared latent $\boldsymbol{z}$ using Eq. (7) and the Wasserstein barycenter (Eq. (10))
29: Generate missing-view latent representations $\bar{\boldsymbol{h}}_0^v$ via reverse diffusion conditioned on shared latent $\boldsymbol{z}$ and randomly sampled view-specific factor $\boldsymbol{w}^v \sim \mathcal{N}(0, \mathbf{I})$
30: Construct complete latent representations $\tilde{\boldsymbol{h}}_0^v$ by combining observed latent features $\boldsymbol{h}_0^v$ and generated missing-view representations $\bar{\boldsymbol{h}}_0^v$
31: Compute shared latent $\boldsymbol{z}$ using Eq. (10) based on the complete latent representations $\tilde{\boldsymbol{h}}_0^v$.
32: Apply k-means clustering in representation $\boldsymbol{z}$ to obtain cluster labels $\boldsymbol{y}$

---

# B. Supplementary Experimental Results

## B.1. Dataset Descriptions

**CUB** (Wah et al., 2011) is a multi-view fine-grained bird dataset across 10 categories, with 600 samples in total and two views: a visual view consisting of 1024-dimensional deep features extracted from bird images, and a textual view consisting of 300-dimensional doc2vec representations derived from bird attribute descriptions. **HandWritten** (van Breukelen et al., 1998) contains handwritten digits '0' to '9', with 2,000 samples in total, and is represented by six views: profile correlations, Fourier coefficients, Karhunen–Loeve coefficients, morphological features, pixel features, and Zernike moments. **LandUse_21** (Yang & Newsam, 2010) is a remote sensing image dataset with 2,100 samples from 21 land-use

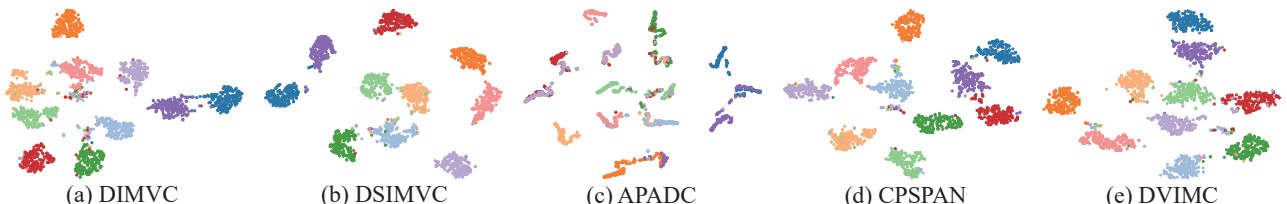

| (a) DIMVC | (b) DSIMVC | (c) APADC | (d) CPSPAN | (e) DVIMC |

*Figure 4.* t-SNE visualization of the learned latent representations on the HandWritten dataset with a missing rate of 0.5.

scene categories, represented by three views corresponding to different handcrafted visual features extracted from satellite images, including spectral-based features, texture descriptors, and structural features. **CCV** (Jiang et al., 2011) is a consumer video dataset containing 6,773 videos from 20 semantic categories, represented by three views constructed from different modalities of video content: spatio-temporal motion features, appearance-based visual features, and audio-related features, each encoded as a 20-dimensional representation. **Reuters** (Lewis et al., 2004) is a multi-view text dataset consisting of 18,758 documents from 6 topic categories, where each document is represented by five views corresponding to the same content written in German, English, French, Italian, and Spanish. **CARL** (Espinosa-Duró, 2013) is a multi-view facial dataset consisting of 2,460 paired samples, where each sample contains three modalities: grayscale, infrared, and thermal images.

### B.2. Implementation Details

For each dataset, clustering performance is evaluated under four different missing rates, denoted by $\eta \in \{0.1, 0.3, 0.5, 0.7\}$. To construct incomplete multi-view datasets, given a dataset with $N$ samples and $V$ views, we randomly select $N \times \eta$ samples and apply missing-view corruption. For each selected sample, at least one view is randomly removed while ensuring that at least one view remains available. All experiments are conducted on a local server equipped with an NVIDIA GeForce RTX 4090 Ti GPU and 64 GB RAM, running Ubuntu 18.04. For all datasets, the dimensionality of the view-specific latent representation $\boldsymbol{h}^v$ is set to 64, the shared semantic latent representation $\boldsymbol{z}^v$ is set to 64, and the auxiliary view-specific latent variable $\boldsymbol{w}^v$ is set to 32. The number of Gaussian mixture components $\boldsymbol{c}$ is set equal to the ground-truth number of classes for each dataset. The number of diffusion steps $T$ is set to 1000, and the noise schedule $\alpha_t$ is linearly interpolated within the range $\left[1 - e^{-2}, 1 - e^{-5}\right]$. The batch size is fixed to 256. We adopt the Adam optimizer for model training, with the learning rate set to 0.001 during the pretraining stage and 0.0001 during the main training stage. A StepLR scheduler is employed, where the learning rate is decayed by a factor of 0.9 every 10 epochs. The model is pretrained for 100 epochs and subsequently trained for 50 epochs. The hyperparameters $\alpha$ and $\beta$ are selected from the set $\{0.1, 1, 10\}$, while $\gamma$ is fixed to 0.1 for all experiments.

### B.3. Ablation Study

To further investigate the contributions of individual components in the proposed model, we conduct a comprehensive ablation study under a missing rate of 0.1. Specifically, we consider three variants: (1) w/o MI, which removes the mutual information (MI) constraint; (2) w/o Diff, which replaces the diffusion-based generative module with a simple multilayer perceptron (MLP); and (3) w/o MI & Diff, which excludes both the MI constraint and the diffusion model. The results are reported in Tables 4. As shown in the table, removing any key component consistently leads to performance degradation across all datasets, demonstrating that each module plays an important role in the overall framework. In particular, eliminating the MI constraint results in a substantial drop in performance, especially on datasets such as CUB and HandWritten. This indicates that the MI constraint is crucial for enforcing effective disentanglement, ensuring that the shared latent variables capture meaningful cross-view semantic information. Replacing the diffusion-based generator with a standard MLP also leads to noticeable performance degradation. This performance drop suggests that the MLP lacks sufficient expressive power to accurately model complex data distributions, resulting in less reliable view imputation and increased reconstruction errors. Consequently, the quality of the completed views deteriorates, negatively affecting clustering performance. When both the MI constraint and the diffusion module are removed, the model exhibits further performance degradation, confirming the complementary nature of these two components. Overall, the ablation results clearly demonstrate that the diffusion-based generation mechanism and the MI-driven disentanglement constraint jointly contribute to the superior performance of the proposed method in incomplete multi-view clustering.

In addition, we further evaluate the effectiveness of the proposed mixture MWB by comparing it with mean aggregation and PoE under a missing rate of 0.1. Mean aggregation performs simple averaging and may blur modality-specific structures, while PoE tends to produce over-confident estimates by shrinking uncertainty. As shown in Table 5, MWB achieves superior performance on most datasets, particularly on HandWritten and CCV, and consistently obtains competitive results across different evaluation metrics, demonstrating the effectiveness of Wasserstein-based latent aggregation for robust cross-view alignment.

*Table 4.* Ablation study of different model variants under a missing rate of 0.1.

| Dataset | Method | ACC | NMI | ARI |
|---|---|---|---|---|
| CUB | IDCD | 82.00 | 79.31 | 69.21 |
| | w/o MI | 38.00 | 27.27 | 15.26 |
| | w/o Diff | 80.17 | 77.57 | 67.73 |
| | w/o MI & Diff | 74.33 | 75.23 | 61.66 |
| HandWritten | IDCD | 97.17 | 93.56 | 93.83 |
| | w/o MI | 71.80 | 68.43 | 57.28 |
| | w/o Diff | 94.25 | 90.70 | 90.10 |
| | w/o MI & Diff | 91.25 | 88.84 | 88.96 |
| LandUse_21 | IDCD | 27.90 | 35.66 | 15.08 |
| | w/o MI | 14.22 | 10.79 | 3.49 |
| | w/o Diff | 25.10 | 31.15 | 12.13 |
| | w/o MI & Diff | 17.25 | 14.91 | 4.97 |
| CCV | IDCD | 24.15 | 21.62 | 9.22 |
| | w/o MI | 11.56 | 9.02 | 2.17 |
| | w/o Diff | 21.26 | 20.19 | 8.32 |
| | w/o MI & Diff | 17.11 | 14.84 | 5.88 |
| Reuters | IDCD | 53.46 | 35.53 | 29.23 |
| | w/o MI | 50.77 | 30.86 | 25.93 |
| | w/o Diff | 51.57 | 31.68 | 26.65 |
| | w/o MI & Diff | 49.45 | 29.24 | 25.02 |

*Table 5.* Comparison of different latent aggregation strategies under a missing rate of 0.1.

| Dataset | Method | ACC | NMI | ARI |
|---|---|---|---|---|
| CUB | Mean | 81.17 | 79.02 | 67.71 |
| | PoE | 71.00 | 75.33 | 61.43 |
| | MWB | 82.00 | 79.31 | 69.21 |
| HandWritten | Mean | 95.65 | 91.20 | 90.58 |
| | PoE | 95.75 | 91.39 | 90.78 |
| | MWB | 97.17 | 93.56 | 93.83 |
| LandUse_21 | Mean | 26.90 | 34.38 | 14.49 |
| | PoE | 27.38 | 34.14 | 14.75 |
| | MWB | 27.90 | 35.66 | 15.08 |
| CCV | Mean | 22.90 | 19.72 | 8.49 |
| | PoE | 21.70 | 18.40 | 7.62 |
| | MWB | 24.15 | 21.62 | 9.22 |
| Reuters | Mean | 53.05 | 33.63 | 28.99 |
| | PoE | 52.62 | 32.74 | 27.23 |
| | MWB | 53.46 | 35.53 | 29.23 |

## B.4. Parameter Analysis

We conduct a sensitivity analysis on the hyperparameters $\alpha$ and $\beta$ to investigate their impact on clustering performance. Specifically, both parameters are varied within the set $\{0.01, 0.1, 1, 10, 100\}$. The resulting ACC under different parameter

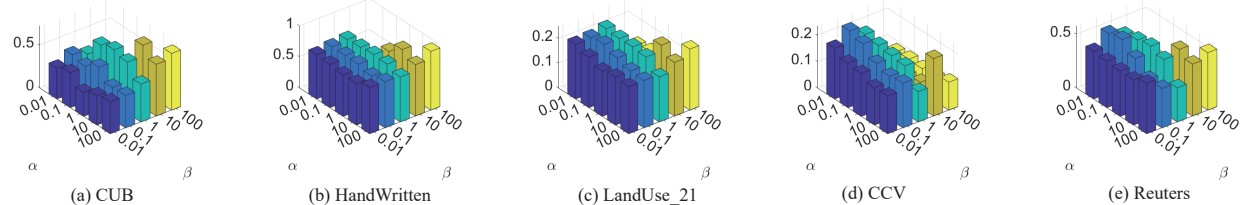

*Figure 5.* Sensitivity analysis of hyperparameters $\alpha$ and $\beta$ on ACC across five datasets with a missing rate of 0.7.

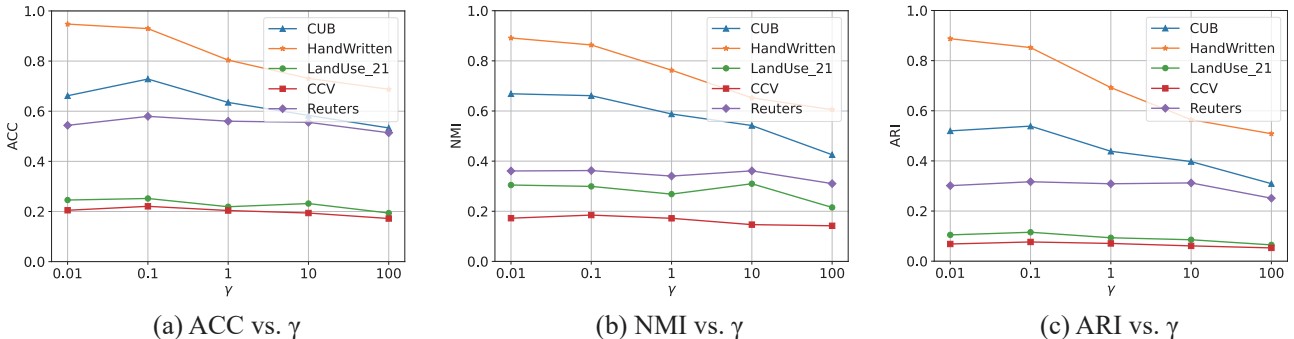

*Figure 6.* Sensitivity analysis of hyperparameter $\gamma$ across five datasets with a missing rate of 0.7.

configurations is illustrated in Figure 5. As observed from the results, the model achieves consistently better performance when $\alpha$ and $\beta$ are of comparable magnitude, whereas a significant performance degradation occurs when the two parameters differ substantially in scale. This phenomenon is well aligned with the design philosophy of our model. The hyperparameters $\alpha$ and $\beta$ jointly regulate the balance between view reconstruction and the disentanglement of shared and view-specific representations. When the two parameters are properly balanced, the model can effectively learn disentangled latent representations while preserving essential information from each view. In contrast, an excessively large $\alpha$ places disproportionate emphasis on view recovery, which weakens the separation between shared and view-specific representations. Conversely, an overly large $\beta$ enforces overly strong disentanglement constraints, potentially compromising the fidelity of the recovered representations and leading to information loss. Consequently, both scenarios result in suboptimal clustering performance. Based on these observations, we recommend setting $\alpha$ and $\beta$ to values within the same order of magnitude to ensure a favorable trade-off between representation disentanglement and information preservation.

We further investigate the sensitivity of the proposed model to the hyperparameter $\gamma$, which regulates the strength of cross-view alignment among shared latent representations. As shown in Figure 6, we evaluate the clustering performance under a missing rate of 0.1 on all five datasets while varying the value of $\gamma$. The results reveal that overly large values of $\gamma$ lead to a clear performance degradation. This can be attributed to the fact that $\gamma$ controls the relative importance of enforcing cross-view consistency. When $\gamma$ is excessively large, the optimization process becomes dominated by the alignment objective, causing the model to favor overly constrained shared representations that are less adaptable to the intrinsic data structure. As a result, the learned representations may deviate from the optimal clustering configuration. In contrast, moderate values of $\gamma$ strike a better balance between cross-view alignment and representation flexibility, yielding more stable and superior clustering performance. Based on these observations, we set $\gamma$ to 0.1 for all experiments.

In addition, we conduct a sensitivity analysis on the diffusion step $T$ under a missing rate of 0.7 to investigate its impact on clustering performance. Specifically, $T$ is varied from 500 to 2000. The corresponding results are reported in Figure 7. As observed, the proposed method maintains relatively stable performance across a broad range of diffusion steps, indicating low sensitivity to the choice of $T$. The best performance is generally achieved at $T = 1000$. When $T$ is too small, the diffusion process may be insufficient to adequately model complex data distributions, whereas excessively large values of $T$ may introduce redundant noise and result in slight performance degradation. Based on these observations, we set $T = 1000$ in all experiments to achieve a favorable balance between effectiveness and robustness.

Furthermore, we evaluate the sensitivity of the proposed method to the number of mixture components $K$ under a missing rate of 0.7. Specifically, we vary $K$ across multiple datasets, and the corresponding results are summarized in Table 6. As

observed, the proposed method remains relatively robust to the choice of $K$, with only moderate performance variations when $K$ deviates from the ground-truth number of clusters. For instance, the best performance on the CUB dataset is achieved at $K = 10$, while other settings result in only slight performance degradation. On the HandWritten dataset, the performance remains stable across all tested values of $K$. Similar trends can also be observed on other datasets, indicating that the proposed method is relatively insensitive to cluster number misspecification and maintains stable clustering performance under different settings of $K$.

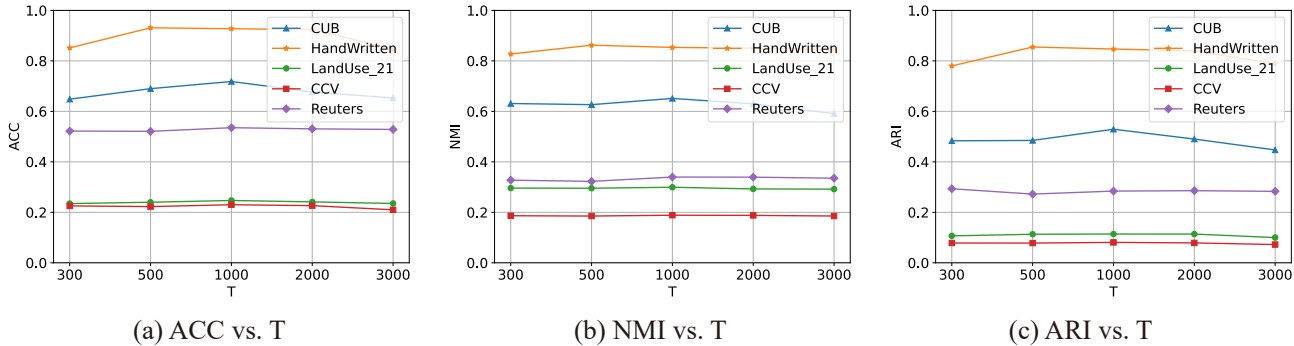

| (a) ACC vs. T | (b) NMI vs. T | (c) ARI vs. T |

*Figure 7.* Sensitivity analysis of hyperparameter $T$ across five datasets with a missing rate of 0.7.

*Table 6.* Sensitivity analysis of the number of clustering components $K$ under a missing rate of 0.7.

| Dataset | $K$ | ACC | NMI | ARI |
|---|---|---|---|---|
| CUB | 5 | 49.83 | 52.12 | 32.21 |
| | 10 | 71.83 | 65.13 | 52.92 |
| | 15 | 69.83 | 64.33 | 50.13 |
| | 20 | 65.00 | 61.47 | 46.41 |
| | 25 | 60.17 | 58.79 | 43.76 |
| HandWritten | 5 | 91.70 | 85.27 | 82.92 |
| | 10 | 92.73 | 85.34 | 84.69 |
| | 15 | 92.55 | 85.94 | 83.75 |
| | 20 | 92.40 | 85.73 | 84.00 |
| | 25 | 92.00 | 85.32 | 83.20 |
| LandUse_21 | 5 | 23.19 | 27.52 | 10.16 |
| | 10 | 21.81 | 27.47 | 9.32 |
| | 15 | 21.38 | 27.47 | 8.93 |
| | 20 | 24.01 | 29.27 | 11.39 |
| | 21 | 24.70 | 29.94 | 11.41 |
| | 25 | 24.14 | 29.42 | 10.71 |
| CCV | 5 | 20.52 | 17.83 | 7.31 |
| | 10 | 21.44 | 17.52 | 7.23 |
| | 15 | 22.06 | 18.48 | 7.67 |
| | 20 | 23.01 | 18.86 | 8.09 |
| | 25 | 24.38 | 19.36 | 8.81 |
| Reuters | 5 | 56.03 | 36.30 | 31.55 |
| | 6 | 53.53 | 33.98 | 28.42 |
| | 10 | 51.10 | 32.04 | 27.54 |
| | 15 | 51.91 | 34.39 | 28.77 |
| | 20 | 52.60 | 34.37 | 28.63 |
| | 25 | 52.77 | 34.14 | 28.46 |

## B.5. Disentanglement Analysis

*Table 7.* Disentanglement analysis under a missing rate of 0.1, including HSIC-based independence evaluation and clustering performance using shared and view-specific latent representations.

| Dataset | representation | ACC | NMI | ARI | HSIC |
|---------|---------------|-----|-----|-----|------|
| CUB | $w^1$ | 16.33 | 3.75 | 0.32 | 0.0553 |
| | $w^2$ | 15.17 | 2.98 | -0.17 | 0.057097 |
| | $w_{\text{concat}}$ | 15.67 | 2.64 | -0.14 | 0.077504 |
| | $z$ | 82.00 | 79.31 | 69.21 | |
| HandWritten | $w^1$ | 12.85 | 0.89 | -0.01 | 0.014664 |
| | $w^2$ | 12.70 | 0.73 | -0.07 | 0.014759 |
| | $w^3$ | 12.90 | 0.90 | -0.01 | 0.01512 |
| | $w^4$ | 13.10 | 0.96 | 0.05 | 0.014117 |
| | $w^5$ | 13.05 | 0.75 | -0.07 | 0.014594 |
| | $w^6$ | 12.60 | 0.88 | 0.01 | 0.021153 |
| | $w_{\text{concat}}$ | 13.25 | 0.94 | 0.02 | 0.037216 |
| | $z$ | 97.17 | 93.56 | 93.83 | |
| LandUse_21 | $w^1$ | 8.76 | 3.71 | 0.09 | 0.017096 |
| | $w^2$ | 8.81 | 3.52 | 0.05 | 0.017998 |
| | $w^3$ | 8.57 | 3.38 | 0.03 | 0.015871 |
| | $w_{\text{concat}}$ | 8.90 | 3.53 | 0.09 | 0.029001 |
| | $z$ | 27.90 | 35.66 | 15.08 | |
| CCV | $w^1$ | 14.78 | 10.45 | 3.83 | 0.110847 |
| | $w^2$ | 14.75 | 8.48 | 2.71 | 0.109854 |
| | $w^3$ | 13.45 | 8.64 | 2.51 | 0.083072 |
| | $w_{\text{concat}}$ | 18.69 | 13.60 | 5.46 | 0.112135 |
| | $z$ | 24.15 | 21.62 | 9.22 | |
| Reuters | $w^1$ | 17.49 | 0.04 | 0.04 | 0.002307 |
| | $w^2$ | 18.55 | 0.19 | 0.12 | 0.008971 |
| | $w^3$ | 18.72 | 0.29 | 0.23 | 0.011184 |
| | $w^4$ | 18.18 | 0.12 | 0.04 | 0.005534 |
| | $w^5$ | 18.75 | 0.22 | 0.13 | 0.007588 |
| | $w_{\text{concat}}$ | 18.44 | 0.37 | 0.29 | 0.015822 |
| | $z$ | 53.46 | 35.53 | 29.23 | |

## B.6. Qualitative Examples of Reconstruction Results

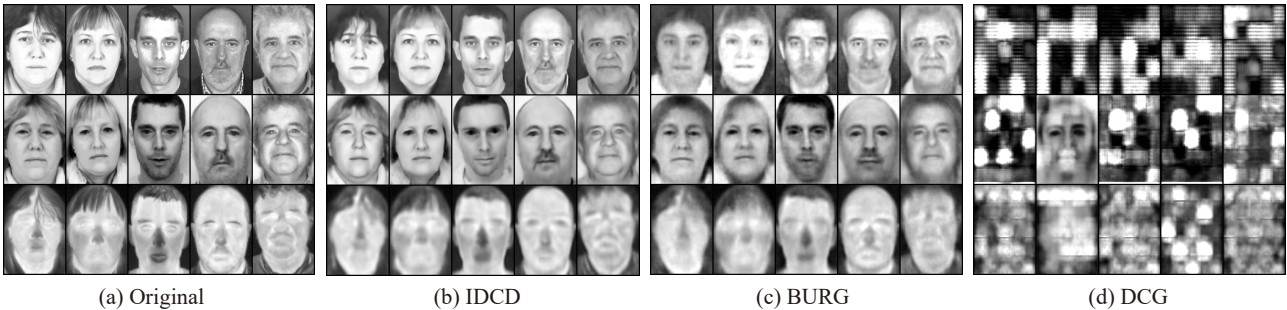

(a) Original      (b) IDCD      (c) BURG      (d) DCG

*Figure 8.* Qualitative comparison of reconstructed samples on the CARL dataset under a missing rate of 0.1.

