# OpenReview forum: "Information-Theoretic Disentangled Latent Modeling with Conditional Diffusion for Incomplete Multi-View Clustering"
_ICML.cc/2026/Conference — ICML 2026 spotlight_

### Official Review · Reviewer_HoEt · 2026-02-25

**Soundness:** 3
**Presentation:** 3
**Significance:** 3
**Originality:** 3
**Overall Recommendation:** 5
**Confidence:** 5

**Summary:**

This paper addresses the problem of incomplete multi-view clustering by proposing an information-theoretic conditional diffusion–based generative framework. The model integrates conditional diffusion into a variational inference framework to generate and complete missing views, thereby improving clustering performance under incomplete observations. Moreover, the method introduces information-theoretic objectives to disentangle shared semantic information from view-specific factors in the latent space, enabling more accurate capture of common semantics while mitigating interference from private information. The paper is supported by careful mathematical derivations, including the formulation of an evidence lower bound (ELBO), and presents a well-structured model architecture. Extensive experiments conducted on multiple datasets under various missing-view ratios demonstrate the effectiveness of the proposed approach.

**Compliance With Llm Reviewing Policy:**

Affirmed.

**Key Questions For Authors:**

1. The paper adopts a mixture Wasserstein barycenter to aggregate semantic posteriors across views. The authors should clarify whether this choice is essential in practice and explain how it compares, conceptually or empirically, to simpler fusion strategies such as averaging or product-of-experts. The authors should also articulate the specific advantages provided by Wasserstein-based aggregation in this setting.
2. The paper introduces disentanglement objectives based on mutual information. The authors should clarify the intended notion of disentanglement in this work, specifically whether the objective aims to achieve strict identifiability of latent factors or a weaker, functional separation.
3. The model relies on conditional diffusion for generative imputation, but the procedure for generating missing views is not clearly specified. The authors should clarify whether the random subset sampling strategy used during training is also applied during inference, or whether a fixed subset of views is employed.

**Limitations:**

The paper does not explicitly discuss its limitations. Adding a short limitations section that outlines key modeling assumptions, practical constraints, and scenarios where the method may be less effective would improve transparency and reader understanding.

**Strengths And Weaknesses:**

1. The paper combines information-theoretic principles and conditional diffusion models within a unified variational framework, and derives a corresponding evidence lower bound.
2. The use of conditional diffusion to generate missing views provides a more principled and realistic imputation strategy compared to no-imputation or random-filling baselines, leading to improved handling of incomplete data.
3. By leveraging mutual-information-based objectives to disentangle shared and view-specific latent representations, the method enhances the extraction of shared semantic information and reduces interference from private factors.
4. The experimental evaluation is thorough, covering multiple datasets and missing rates, and is complemented by sufficient theoretical analysis.
Weaknesses
1. The paper employs a mixture Wasserstein barycenter to aggregate semantic posteriors across views. While this choice is theoretically motivated, it remains unclear whether it is essential in practice. A comparison with, or discussion of, simpler fusion strategies (e.g., averaging or product-of-experts) would help assess the practical benefits of Wasserstein-based aggregation.
2. While the paper introduces disentanglement objectives based on mutual information, the precise meaning of disentanglement remains ambiguous. In particular, it is unclear whether the goal is strict identifiability or a weaker, functional separation of latent factors. Clarifying this distinction would strengthen the theoretical interpretation of the proposed method.
3. The model relies on generative imputation via conditional diffusion; however, the paper does not clearly explain how missing views are generated in practice. In particular, while the training procedure involves randomly sampling subsets of observed views to learn the fused shared representation, it is unclear whether the same random subset strategy is used during the inference stage, or whether a fixed subset of views is employed. Clarifying this distinction would help improve the reproducibility and interpretability of the proposed method.
4. Minor presentation issues. There are some minor labeling errors in the paper. For example, line 198 refers to Appendix A.1, which should instead be Appendix A.2.

---

> ### Author Rebuttal · Authors · 2026-03-31
>
> Q1:While the use of mixture Wasserstein barycenter (MWB) is not strictly required, it provides a more principled way to aggregate modality-specific posteriors by aligning distributions in the Wasserstein space. In contrast, mean aggregation performs simple averaging and may blur modality-specific structures, while product-of-experts (PoE) tends to produce over-confident estimates by shrinking uncertainty.
> Empirically, we provided an ablation comparison under a missing rate of 0.1, where MWB is compared with mean aggregation and PoE. As shown below, MWB consistently achieves competitive or superior performance across datasets and evaluation metrics (ACC, NMI, ARI), demonstrating its practical benefits:
> |Dataset|Method|ACC|NMI|ARI|
> |-|-|-|-|-|
> |CUB|Mean|81.17|79.02|67.71|
> ||PoE|71|75.33|61.43|
> ||MWB|82|79.31|69.21|
> |HandWritten|Mean|95.65|91.2|90.58|
> ||PoE|95.75|91.39|90.78|
> ||MWB|97.17|93.56|93.83|
> |LandUse_21|Mean|26.9|34.38|14.49|
> ||PoE|27.38|34.14|14.75|
> ||MWB|27.9|35.66|15.08|
> |CCV|Mean|22.9|19.72|8.49|
> ||PoE|21.7|18.4|7.62|
> ||MWB|24.15|21.62|9.22|
> |Reuters|Mean|53.05|33.63|28.99|
> ||PoE|52.62|32.74|27.23|
> ||MWB|53.46|35.53|29.23|
>
> These results indicate that MWB better preserves the geometric structure of latent distributions and leads to more robust cross-view alignment. We will clarify both the conceptual advantages and empirical comparisons in the revised manuscript.
>
> Q2:In this work, we do not aim for strict identifiability of latent factors, which typically requires strong assumptions. Instead, we focus on a weaker notion of disentanglement, namely functional separation, where the shared latent captures cross-view semantic information and the view-specific latent encodes modality-specific variations. The mutual information objectives are designed to encourage this separation. We will clarify this distinction in the revised manuscript.
>
> Q3:During training, we randomly sample subsets of observed views to simulate missingness and improve robustness. At inference time, we instead use all available (non-missing) views to infer the shared latent representation, which is then used for conditional diffusion-based generation of missing views. We will clarify this distinction in the revised manuscript.
>
> Minor presentation issues：We will correct the labeling error and carefully check all references in the revised manuscript.
>
> Limitation:Our framework has several limitations. First, it treats all modalities equally during aggregation, which may overlook the imbalance in information quality across views. Incorporating adaptive weighting strategies is a promising direction for future work. Second, the conditional diffusion model introduces additional computational overhead due to iterative denoising. Finally, the generalization ability could be further improved by training on larger and more diverse multimodal datasets.

---

### Official Review · Reviewer_Gdk8 · 2026-03-10

**Soundness:** 4
**Presentation:** 3
**Significance:** 4
**Originality:** 3
**Overall Recommendation:** 5
**Confidence:** 4

**Summary:**

This paper proposes an information-theoretic conditional diffusion framework for incomplete multi-view clustering. The method integrates conditional diffusion models into a variational latent-variable framework to perform principled missing-view imputation, while simultaneously disentangling shared semantic representations from view-specific factors. By explicitly modeling shared and private information in the latent space, the proposed approach aims to improve the robustness and accuracy of clustering under arbitrary missing-view patterns. Beyond the modeling contribution, the paper offers a coherent probabilistic formulation supported by detailed derivations and a clear training objective. The proposed approach is evaluated under a wide range of missing-view settings and demonstrates consistent improvements over existing methods. Overall, the work presents a comprehensive and well-motivated attempt to bridge incomplete multi-view clustering, and constitutes a meaningful contribution to this research area.

**Compliance With Llm Reviewing Policy:**

Affirmed.

**Final Justification:**

This paper proposes an information-theoretic conditional diffusion framework for incomplete multi-view clustering, which presents a comprehensive and well-motivated attempt to bridge incomplete multi-view clustering. In the rebuttal phase, all my concerns have been addressed. I believe this is a solid paper that will contribute to the incomplete multi-view deep learning community. Therefore, I recommend a Accept decision.

**Key Questions For Authors:**

The main questions for the authors are covered by the weaknesses discussed above, and no additional questions are raised.

**Limitations:**

The paper would benefit from a clearer discussion of the scope under which the proposed method is expected to perform well, as well as an analysis of scenarios in which it may fail. Explicitly articulating these limitations in the paper would improve transparency and help readers better assess the practical applicability of the approach.

**Strengths And Weaknesses:**

Strengths
1. The paper presents a unified framework that jointly handles representation learning, missing-view imputation, and clustering, avoiding multi-stage or heuristic pipelines.
2. The integration of conditional diffusion into a variational framework is technically nontrivial and represents a novel modeling direction for incomplete multi-view clustering.
3. The mathematical formulation is detailed and carefully derived, lending theoretical grounding to the proposed approach.
4. Experimental results are extensive and demonstrate consistent improvements over competitive baselines across datasets and missing rates.
Weaknesses
1. The paper mentions the learning rate used during pre-training, but it is unclear which component is being pre-trained. In particular, is the pre-training applied to the encoder Ev, or to another part of the model? Clarifying this point would improve the transparency of the training procedure.
2. While the paper includes implementation details, several aspects of the training procedure are not fully specified, which may hinder exact replication of the results. Providing the implementation code would help readers better understand the overall framework and facilitate reproducibility.
3. The clustering component relies on a predefined number of mixture components in the latent space. The paper does not discuss how sensitive the method is to misspecification of this number, or how performance degrades when the assumed cluster count does not match the true underlying structure.
4. Although the experimental results are strong on average, the paper does not analyze failure cases or scenarios in which the method performs poorly. Such analysis would provide valuable insight into the limitations of the proposed approach and help better assess its practical applicability.
5. The paper presents a unified generative framework for incomplete multi-view clustering, but the distinction between this approach and existing generative or imputation-based IMVC methods could be articulated more clearly.

---

> ### Author Rebuttal · Authors · 2026-03-31
>
> W1:Pre-training is applied to the entire framework rather than a single component (e.g., $E^v$). Using paired data, we jointly pre-train encoders, decoders and diffusion modules to initialize the model parameters. We will clarifiy this procedure.
>
> W2:We will release the code and further clarify the training details to improve reproducibility.
>
> W3:To assess sensitivity to the number of mixture components, we conducted additional experiments by varying the number of clustering components K across multiple datasets, as summarized in the table(missing rate = 0.7). The results indicate that our method is robust to the choice of K, with only mild performance variations when K deviates from the true value. For example, on CUB, the best performance is achieved at K=10, while other values lead to moderate decreases; on HandWritten, performance remains stable across all tested K. Similar trends are observed on other datasets, demonstrating that our method is relatively insensitive to cluster number misspecification.
> |Dataset|K|ACC|NMI|ARI|
> |-|-|-|-|-|
> |CUB|5|49.83|52.12|32.21|
> ||10|71.83|65.13|52.92|
> ||15|69.83|64.33|50.13|
> ||20|65|61.47|46.41|
> ||25|60.17|58.79|43.76|
> |HandWritten|5|91.7|85.27|82.92|
> ||10|92.73|85.34|84.69|
> ||15|92.55|85.94|83.75|
> ||20|92.4|85.73|84|
> ||25|92|85.32|83.2|
> |LandUse_21|5|23.19|27.52|10.16|
> ||10|21.81|27.47|9.32|
> ||15|21.38|27.47|8.93|
> ||20|24.01|29.27|11.39|
> ||21|24.7|29.94|11.41|
> ||25|24.14|29.42|10.71|
> |CCV|5|20.52|17.83|7.31|
> ||10|21.44|17.52|7.23|
> ||15|22.06|18.48|7.67|
> ||20|23.01|18.86|8.09|
> ||25|24.38|19.36|8.81|
> |Reuters|5|56.03|36.3|31.55|
> ||6|53.53|33.98|28.42|
> ||10|51.1|32.04|27.54|
> ||15|51.91|34.39|28.77|
> ||20|52.6|34.37|28.63|
> ||25|52.77|34.14|28.46|
>
> W4:While results are strong overall, we acknowledge failure analysis is important. At a low missing rate(0.1), IDCD is close to the best methods with small gaps in NMI(e.g., Reuters: 35.53 vs. 35.56;LandUse_21: 35.66 vs. 35.93;CCV: 21.62 vs. 22.22), likely due to sufficient observations. At a medium missing rate(0.5), DSIMVC slightly outperforms IDCD on Reuters in NMI(36.85 vs. 35.18), but the gain is modest and it does not address consistent missing-view generation. At a high missing rate(0.7), CPSPAN slightly outperforms IDCD on LandUse_21 in NMI(30.71 vs. 29.94), likely due to stronger robustness under sparse observations, while IDCD’s latent-variable-based generation may be less stable.
>
> W5:Imputation-based methods can be categorized by how they model cross-view relationships and perform completion. Shallow reconstruction methods (e.g., kernel-, graph-, and matrix factorization-based [1,2]) rely on low-rank or local structures but fail to capture complex nonlinear semantics. Structure-aware methods [3,4] leverage prototypes or graphs, yet depend on sufficiently observed samples and degrade under high missing rates. GANs and MI-based methods [5,6] learn cross-view mappings but often require strong supervision and yield entangled representations. Diffusion-based methods [7–9] provide strong generation quality but lack explicit constraints on clustering structure and semantic consistency. In contrast, IDCD disentangles shared and view-specific factors, enforces semantic consistency via information-theoretic constraints and integrates conditional diffusion with clustering, enabling robust joint learning under missing views.
>
> [1] Trivedi A., et al. Multiview clustering with incomplete views, NeurIPS workshop. 2010.
>
> [2] Wan X., et al. Fast continual multi-view clustering with incomplete views, IEEE Transactions on Image Processing, 2024.
>
> [3] Wen J.,  et al. Adaptive graph completion based incomplete multi-view clustering. IEEE Transactions on Multimedia, 2020.
>
> [4] Hong R., et al. Prototype imputation guided incomplete multi-view clustering. IEEE signal processing letters, 2025.
>
> [5] Wang Q., et al. Generative partial multi-view clustering with adaptive fusion and cycle consistency. IEEE Transactions on Image Processing, 2021.
>
> [6] Lin Y., et al. Dual contrastive prediction for incomplete multi-view representation learning. IEEE Transactions on Pattern Analysis and Machine Intelligence, 2023.
>
> [7] Wen J., et al. Diffusion-based Missing-view Generation With the application on incomplete multi-view clustering. ICML. 2024.
>
> [8] Palumbo E., et al. Deep generative clustering with multimodal diffusion variational autoencoders. ICLR. 2024.
>
> [9] Zhang Y., et al. Incomplete multi-view clustering via diffusion contrastive generation. AAAI. 2025.
>
> Limitation: Our framework has several limitations. First, it treats all modalities equally during aggregation, which may overlook the imbalance in information quality across views. Incorporating adaptive weighting strategies is a promising direction for future work. Second, the conditional diffusion model introduces additional computational overhead due to iterative denoising. Finally, the generalization ability could be further improved by training on larger and more diverse multimodal datasets.

---

> > ### Author Rebuttal · Reviewer_Gdk8 · 2026-04-03
> >
> > I thank the authors for their responses. All my concerns have been addressed,and thus I will retain my original score.

---

### Official Review · Reviewer_CMAJ · 2026-03-11

**Soundness:** 4
**Presentation:** 3
**Significance:** 3
**Originality:** 4
**Overall Recommendation:** 5
**Confidence:** 4

**Summary:**

The authors propose IDCD, a framework for incomplete multi-view clustering that combines disentangled latent modeling with conditional diffusion. The method separates each view into shared and view-specific components, enforces semantic consistency through information-theoretic objectives, and performs clustering in a unified latent space with a Gaussian mixture prior. Experiments on five benchmark datasets with varying missing rates show consistent improvements over strong baselines in both clustering performance and missing-view recovery. This is a well-motivated and technically solid contribution to incomplete multi-view learning.

**Compliance With Llm Reviewing Policy:**

Affirmed.

**Final Justification:**

The authors have addressed my concerns after I reviewed their rebuttal.

**Key Questions For Authors:**

1. The structural differences between $z^v$ and $w^v$ are unclear in Figure 1, as they appear visually identical despite differing in the mathematical formulation. The visualization (e.g., color scheme across views) could be improved for better distinguishability.
2. In particular, the manuscript does not provide pseudocode, and the sampling process lacks sufficient detail for full reproducibility.
3. The claim that Wasserstein barycenter aggregation is superior to alternatives such as mean aggregation and product-of-experts is not directly supported by explicit ablation comparisons.
4. The diffusion step is fixed at T=1000 without analysis of its sensitivity. The impact of varying T on performance remains unclear.
5. Given the large number of symbols introduced, the manuscript would benefit from a consolidated notation table to improve clarity.

**Limitations:**

No. The manuscript does not explicitly discuss the limitations of the proposed method. Adding a dedicated section outlining the method’s limitations, such as its reliance on specific assumptions or sensitivity to hyperparameters, would improve the completeness and transparency of the paper.

**Strengths And Weaknesses:**

Strengths
1. The framework tightly integrates disentangled representation learning, information-theoretic objectives, and conditional diffusion in a conceptually consistent manner.
2. The paper is supported by careful mathematical derivations, including a well-formulated evidence lower bound (ELBO), and the mutual information and alignment objectives are clearly developed with appropriate theoretical justification.
3. The use of conditional diffusion for missing-view recovery is well-motivated and naturally complements the clustering objective, enhancing both representation quality and data completion.
4. The method consistently improves clustering and recovery performance, especially as missingness increases, with clearer and more compact clusters than baselines.

Weaknesses
1. The presentation could be further improved. Certain inconsistencies between the illustrative figures and the mathematical formulation may cause confusion and the overall visual clarity could be enhanced. In addition, given the number of symbols introduced, the readability of the manuscript could benefit from better organization of notations.
2. The paper does not provide pseudocode for the overall training framework, and the sampling process is not described in sufficient detail. This limits transparency and may hinder reproducibility.
3. The claimed superiority of Wasserstein barycenter aggregation over alternative aggregation strategies is not directly supported by explicit ablation studies. Stronger empirical evidence would help substantiate this claim.
4. Certain implementation choices, such as fixing the diffusion step T=1000, are not elaborated upon. A sensitivity analysis would help better understand the robustness of the method and the impact of this hyperparameter on performance.

---

> ### Author Rebuttal · Authors · 2026-03-31
>
> Q1:We agree that the distinction between $z^v$ and $w^v$ in Figure 1 is not sufficiently clear. We will improve the visualization with more distinct color schemes and clearer annotations to better differentiate them.
>
> Q2:In the revised manuscript, we will add a complete pseudocode of the proposed method (Algorithm 1) to improve clarity and reproducibility. Specifically, we provide a stage-wise training and inference procedure, including:(i) encoder pretraining, (ii) pretraining on fully paired data, (iii) training on all data (complete & incomplete) and (iv) missing view imputation and inference.
>
> Q3:To better evaluate the effectiveness of MWB, we added an explicit ablation study comparing the proposed Mixture Wasserstein Barycenter (MWB) with Mean aggregation and Product-of-Experts (PoE) under a missing rate of 0.1 across all datasets (as shown in the table below).
> |Dataset|Method|ACC|NMI|ARI|
> |-|-|-|-|-|
> |CUB|Mean|81.17|79.02|67.71|
> ||PoE|71|75.33|61.43|
> ||MWB|82|79.31|69.21|
> |HandWritten|Mean|95.65|91.2|90.58|
> ||PoE|95.75|91.39|90.78|
> ||MWB|97.17|93.56|93.83|
> |LandUse_21|Mean|26.9|34.38|14.49|
> ||PoE|27.38|34.14|14.75|
> ||MWB|27.9|35.66|15.08|
> |CCV|Mean|22.9|19.72|8.49|
> ||PoE|21.7|18.4|7.62|
> ||MWB|24.15|21.62|9.22|
> |Reuters|Mean|53.05|33.63|28.99|
> ||PoE|52.62|32.74|27.23|
> ||MWB|53.46|35.53|29.23|
>
> The results clearly validate our claim. Specifically, MWB consistently achieves better or competitive performance in terms of ACC, NMI and ARI across most datasets. In contrast, Mean aggregation tends to blur modality-specific structures due to naive averaging, while PoE often leads to over-confident posterior estimates, which can degrade clustering performance in several cases. These empirical results, together with the geometric motivation of Wasserstein barycenter in distribution alignment, provide strong evidence supporting the effectiveness of MWB for multi-view latent aggregation. We will further clarify this discussion in the revised manuscript.
>
> Q4:Across all datasets (see table below), we observe that the performance remains relatively stable across a wide range of diffusion steps (T=300–3000) under a missing rate of 0.7. The best results are typically achieved at T=1000. In contrast, smaller values of T may lead to insufficient diffusion modeling, while excessively large values of T can introduce unnecessary noise and result in mild performance degradation. Overall, the proposed method is not highly sensitive to the choice of T within a reasonable range, which justifies fixing T=1000 in all experiments for simplicity and robustness.
> |Dataset|T|ACC|NMI|ARI|
> |-|-|-|-|-|
> |CUB|300|64.83|63.11|48.33|
> ||500|69|62.66|48.49|
> ||1000|71.83|65.13|52.92|
> ||2000|67.67|62.98|49.03|
> ||3000|65.33|59.13|44.71|
> |HandWritten|300|85.15|82.72|78|
> ||500|93.1|86.22|85.5|
> ||1000|92.73|85.34|84.69|
> ||2000|92.4|85.03|83.95|
> ||3000|84.7|84.42|78.88|
> |LandUse_21|300|23.48|29.61|10.68|
> ||500|24|29.55|11.32|
> ||1000|24.7|29.94|11.41|
> ||2000|24.14|29.27|11.39|
> ||3000|23.52|29.19|10.01|
> |CCV|300|22.55|18.66|7.85|
> ||500|22.28|18.53|7.83|
> ||1000|23.01|18.86|8.09|
> ||2000|22.65|18.79|7.89|
> ||3000|21|18.55|7.24|
> |Reuters|300|52.2|32.74|29.34|
> ||500|52.09|32.27|27.23|
> ||1000|53.53|33.98|28.42|
> ||2000|53.07|33.93|28.58|
> ||3000|52.85|33.53|28.31|
>
> Q5:To enhance readability and clarity, we added a consolidated notation table to summarize all key symbols. A portion of the notation table is shown below and the complete version will be provided in the Appendix.
> |Symbol|Meaning|Space|
> |-|-|-|
> |$X^v$|Observed data in view $v$|$\mathbb{R}^{n_v \times d_v}$|
> |$x_i^v$, $x^v$|$i$-th instance in view $v$|$\mathbb{R}^{d_v}$|
> |$g^v$|Binary indicator for observed samples|$\lbrace 0,1 \rbrace^n$|
> |$h_0^v$|Encoded latent of view $v$|$\mathbb{R}^{d_h}$|
> |$h_t^v$|Diffused latent representation at step $t$|$\mathbb{R}^{d_h}$|
> |$z^v$|View-specific semantic latent|$\mathbb{R}^{d}$|
> |$w^v$|View-specific factor|$\mathbb{R}^{d_w}$|
> |$\mu_{w^v}$|Mean of the variational distribution of $w^v$|$\mathbb{R}^{d_w}$|
> |$\sigma_{w^v}$|Standard deviation of the variational distribution of $w^v$|$\mathbb{R}^{d_w}$|
> |$z$|Aggregated shared latent|$\mathbb{R}^{d}$|
> |{$z^v$}|Set of view-specific semantic latent|-|
> |{$w^v$}|Set of view-specific latent factors|-|
> |{$h_{1:T}^v$}|Diffusion trajectories for all views|-|
> |$T$|Number of diffusion steps|$\mathbb{N}_+$|
>
>
> Limitation:Our framework has several limitations. First, it treats all modalities equally during aggregation, which may overlook the imbalance in information quality across views. Incorporating adaptive weighting strategies is a promising direction for future work. Second, the conditional diffusion model introduces additional computational overhead due to iterative denoising. Finally, the generalization ability could be further improved by training on larger and more diverse multimodal datasets.

---

> > ### Author Rebuttal · Reviewer_CMAJ · 2026-04-02
> >
> > The authors address my concerns.

---

### Official Review · Reviewer_4ayi · 2026-03-14

**Soundness:** 3
**Presentation:** 3
**Significance:** 3
**Originality:** 3
**Overall Recommendation:** 5
**Confidence:** 3

**Summary:**

The authors developed an approach to multi-view clustering with missing views. The approach attempts to disentangle global and view-specific information. It uses a diffusion model conditioned on learned disentangled latents to recover missing views in order to improve clustering.

**Compliance With Llm Reviewing Policy:**

Affirmed.

**Final Justification:**

The rebuttal addressed my concerns and provided enough additional results to convince me to raise my score.

**Key Questions For Authors:**

1) I would have liked to see examples of the reconstructed missing views, but I see that most of the datasets are already featurized in some way. Have you tried your approach directly on images or on encodings of images that can be decoded? What about text?
2) From 3.2, "Following the Markov chain $h^v$ → $z^v$ → z, we assume conditional independence between the view-specific latents $z^v$ given the shared latent." Doesn't conditioning on z make $z^v$ dependent in this chain?
3) Which latent representation was used to make the t-sne visualizations?
4) Have you done any empirical studies to verify how disentangled $w^v$ is from z

**Limitations:**

yes

**Strengths And Weaknesses:**

Soundness:
The claims appear to be backed by experiment and theory.

Presentation:
I like the ideas in this paper and the coherent approach developed. But if I had to describe the presentation of this paper in one word, it would be "concise." A little more exposition would have made the methods section easier to parse. Although the block-diagram figure helped a lot, I found it hard to follow the equations in 3.2 at first because the terms weren't all explicitly defined up front. I would have also liked to know up-front what space each variable belongs to, e.g. $w^v \in \mathbb{R}^{d_w}$. Even though it can be inferred pretty easily, things like {$w^v$} should also be explicitly defined.  The indicator $g^v$ is introduced up front, but is it ever used?

Even though the block diagram is very good, I wish it had a more detailed description summarizing the method.

The equation between 7 and 8 has a typo just before it and it should probably get its own line.

---

> ### Author Rebuttal · Authors · 2026-03-31
>
> Weakness:In the revised version, we will expand Section 3.2, clarify all variables and their corresponding spaces at first occurrence and include a notation summary of key variables, as partially shown below:
> |Symbol|Meaning|Space|
> |-|-|-|
> |$X^v$|Observed data in view $v$|$\mathbb{R}^{n_v \times d_v}$|
> |$x_i^v$, $x^v$|$i$-th instance in view $v$|$\mathbb{R}^{d_v}$|
> |$z^v$|View-specific semantic latent|$\mathbb{R}^{d}$|
> |$w^v$|View-specific factor|$\mathbb{R}^{d_w}$|
> |{$w^v$}|Set of view-specific latent factors|-|
>
> Regarding the indicator variable $\mathbf{g}^v$, we will clarify that it denotes view availability and that $\mathbf{g}^{\text{all}} \gets \bigwedge_{v=1}^V \mathbf{g}^v$ indicates samples observed in all views and is used in the pseudo-code. We will also revise the detailed description below the block diagram for clarity and correct the typo between Eq. (7) and Eq. (8) by placing the equation on a separate line.
>
> Q1:In the revised version, we will include qualitative examples of reconstructed samples and quantitative comparison results to illustrate the reconstruction quality of our method. Due to the reply constraints, we only present the quantitative comparison results. Three evaluation metrics are adopted for comprehensive performance comparison, i.e. RMSE, NMSE and PSNR. Specifically, RMSE measures absolute reconstruction error, NMSE normalizes it for scale invariance and PSNR reflects reconstruction fidelity via signal-to-noise ratio. Among all baseline models, we selected two generative-based models BURG and DCG for fair comparisons and conducted experiments on the CARL dataset with a missing rate of 0.1. The results are summarized below. As shown in the table, our method consistently outperformed the other two alternatives across the three views, highlighting the reconstruction ability of our method.
> |View|Method|RMSE↓|NMSE↓|PSNR↑|
> |-|-|-|-|-|
> |CLASSIC|DCG|0.52|0.7|5.75|
> ||BURG|0.2|0.11|13.85|
> ||IDCD|0.19|0.09|14.58|
> |INFRARED|DCG|0.61|0.85|4.34|
> ||BURG|0.21|0.1|13.55|
> ||IDCD|0.19|0.09|14.3|
> |THERMAL|DCG|0.35|0.26|9.16|
> ||BURG|0.16|0.06|15.7|
> ||IDCD|0.14|0.04|17.09|
>
> Q2: The statement in Section 3.2 is inaccurate and will be corrected. In our framework, the Markov chain is only used to imply the inference process and our conditional independence assumption is that each view is processed by an independent encoder $q_\phi(z^v \mid h^v)$, operating solely on its corresponding observation. The factorization in Eq. (8) follows from this independent  encoding design. We will revise the manuscript accordingly.
>
> Q3:The t-SNE visualizations are based on the shared latent $z$, capturing aggregated multi-view semantics, which we will clarify in the revision.
>
> Q4:To empirically evaluate the disentanglement between the shared latent variable $z$ and the view-specific variables $w^v$, we conducted the following two experiments. First, we adopted the HSIC (Hilbert-Schmidt Independence Criterion) metric to measure the statistical dependence between them, since $z$ and $w^v$ generally lie in different dimensional spaces. A lower HSIC value indicates weaker dependence, suggesting better disentanglement. In addition, we also computed HSIC between $z$ and the concatenated representation $w = \mathrm{Concat}([w^v])$ to further evaluate whether aggregated view-specific information remains independent of the shared representation. Second, since the shared latent $z$ contains the unique class semantics shared by all views while $w^v$ only encodes view-specific information, it is expected that the clustering performance using $w^v$ alone will be significantly inferior to those using $z$. Therefore, we further demonstrated the clustering results using each of the $w^v$ and $w$. The obtained ACC, NMI and ARI values shown in the table below confirmed our assumption. In summary, the results show that $z$ captures shared semantics, while $w^v$ and $w$ encode view-specific information. The results under a missing rate of 0.1 are shown below:
> |Dataset|representation|ACC|NMI|ARI|HSIC($\cdot$, $z$)|
> |-|-|-|-|-|-|
> |CUB|$w^1$|16.33|3.75|0.32|0.0553|
> ||$w^2$|15.17|2.98|-0.17|0.0570|
> ||$w$|15.67|2.64|-0.14|0.0775|
> ||$z$|82|79.31|69.21||
> |HandWritten|$w^1$|12.85|0.89|-0.01|0.0146|
> ||$w^2$|12.7|0.73|-0.07|0.0147|
> ||$w^3$|12.9|0.9|-0.01|0.0151|
> ||$w^4$|13.1|0.96|0.05|0.0141|
> ||$w^5$|13.05|0.75|-0.07|0.0145|
> ||$w^6$|12.6|0.88|0.01|0.0211|
> ||$w$|13.25|0.94|0.02|0.0372|
> ||$z$|97.17|93.56|93.83||
> |LandUse_21|$w^1$|8.76|3.71|0.09|0.0170|
> ||$w^2$|8.81|3.52|0.05|0.0179|
> ||$w^3$|8.57|3.38|0.03|0.0158|
> ||$w$|8.9|3.53|0.09|0.0290|
> ||$z$|27.9|35.66|15.08||
> |CCV|$w^1$|14.78|10.45|3.83|0.1108|
> ||$w^2$|14.75|8.48|2.71|0.1098|
> ||$w^3$|13.45|8.64|2.51|0.0830|
> ||$w$|18.69|13.6|5.46|0.1121|
> ||$z$|24.15|21.62|9.22||
> |Reuters|$w^1$|17.49|0.04|0.04|0.0023|
> ||$w^2$|18.55|0.19|0.12|0.0089|
> ||$w^3$|18.72|0.29|0.23|0.0111|
> ||$w^4$|18.18|0.12|0.04|0.0055|
> ||$w^5$|18.75|0.22|0.13|0.0075|
> ||$w$|18.44|0.37|0.29|0.0158|
> ||$z$|53.46|35.53|29.23||

---

> > ### Author Rebuttal · Reviewer_4ayi · 2026-04-03
> >
> > The authors have addressed my questions and concerns. I will adjust my score.

---

### Decision · Program_Chairs · 2026-04-30

**Decision:**

Accept (spotlight)

**Comment:**

This paper proposes a novel information-theoretic conditional diffusion framework (IDCD) for incomplete multi-view clustering. The method integrates conditional diffusion models to generate missing views in a principled manner while using mutual information objectives to separate shared semantic information from view-specific factors, with a Gaussian mixture prior enabling clustering directly in the latent space. Extensive experiments across five benchmark datasets under varying missing rates demonstrate consistent and significant improvements over strong baselines in both clustering accuracy and missing-view recovery. Overall, the proposed model is technically novel, and the evaluation is comprehensive and convincing. All reviewers unanimously recommended acceptance after the rebuttal. Based on their recommendations, the paper is suggested for acceptance.